# Impact of color-coded and warning nutrition labelling schemes: A systematic review and network meta-analysis

**Jing Song**[1☯], **Mhairi K. Brown**[1☯], **Monique Tan**[1], **Graham A. MacGregor**[1],
**Jacqui Webster**[2], **Norm R. C. Campbell**[3], **Kathy Trieu**[2], **Cliona Ni Mhurchu**[2,4], **Laura K. Cobb**[5], **Feng J. He**[1] *

**1** Wolfson Institute of Population Health, Barts and The London School of Medicine & Dentistry, Queen Mary University of London, London, United Kingdom, **2** The George Institute for Global Health, University of New South Wales, Newtown, Australia, **3** Department of Medicine and Libin Cardiovascular Institute, University of Calgary, Alberta, Canada, **4** National Institute for Health Innovation, University of Auckland, Auckland, New Zealand, **5** Resolve to Save Lives, Vital Strategies, New York City, New York, United States of America

☯ These authors contributed equally to this work.
* f.he@qmul.ac.uk

**Data Availability Statement:** All relevant data are contained within the Supporting Information files.

**Funding:** The authors received no specific funding for this work.

## Abstract

### Background

Suboptimal diets are a leading risk factor for death and disability. Nutrition labelling is a potential method to encourage consumers to improve dietary behaviour. This systematic review and network meta-analysis (NMA) summarises evidence on the impact of colour-coded interpretive labels and warning labels on changing consumers' purchasing behaviour.

### Methods and findings

We conducted a literature review of peer-reviewed articles published between 1 January 1990 and 24 May 2021 in PubMed, Embase via Ovid, Cochrane Central Register of Controlled Trials, and SCOPUS. Randomised controlled trials (RCTs) and quasi-experimental studies were included for the primary outcomes (measures of changes in consumers' purchasing and consuming behaviour). A frequentist NMA method was applied to pool the results. A total of 156 studies (including 101 RCTs and 55 non-RCTs) nested in 138 articles were incorporated into the systematic review, of which 134 studies in 120 articles were eligible for meta-analysis. We found that the traffic light labelling system (TLS), nutrient warning (NW), and health warning (HW) were associated with an increased probability of selecting more healthful products (odds ratios [ORs] and 95% confidence intervals [CIs]: TLS, 1.5 [1.2, 1.87]; NW, 3.61 [2.82, 4.63]; HW, 1.65 [1.32, 2.06]). Nutri-Score (NS) and warning labels appeared effective in reducing consumers' probability of selecting less healthful products (NS, 0.66 [0.53, 0.82]; NW,0.65 [0.54, 0.77]; HW,0.64 [0.53, 0.76]). NS and NW were associated with an increased overall healthfulness (healthfulness ratings of products purchased using models such as FSAm-NPS/HCSP) by 7.9% and 26%, respectively. TLS, NS, and NW were associated with a reduced energy (total energy: TLS, −6.5%; NS, −6%; NW,

**Competing interests:** I have read the journal's policy and the authors of this manuscript have the following competing interests: JS is funded, and GAM and FJH are partially funded by the National Institute for Health Research (NIHR) (16/136/77) using UK aid from the UK Government to support global health research. The views expressed in this publication are those of the authors and not necessarily those of the NIHR or the UK government. GAM is the chair of Blood Pressure UK, Action on Salt, Sugar and Health and World Action on Salt, Sugar and Health (WASSH). FJH is a member of Action on Salt, Sugar and Health and WASSH. Blood Pressure UK, Action on Salt, Sugar and Health and WASSH are nonprofit charitable organizations. GAM and FJH do not receive any financial support from them. MKB is funded by the Medical Research Council/UK Research and Innovation under the Newton Fund Impact Scheme call (grant MR/V005847/1) and Vital Strategies. JW is the Director of the WHO Collaborating Centre on Population Salt Reduction, is supported by a National Heart Foundation Career Development Fellowship (1082924), and through an NHMRC Centre of Research Excellence on food policy interventions to reduce salt (1117300). KT is supported by an NHMRC Early Career Fellowship (1161597) and a Postdoctoral Fellowship (102140) from the National Heart Foundation of Australia. NRCC reports personal fees from Resolve to Save Lives (RTSL), and the World Bank outside the submitted work; and an unpaid consultant on dietary sodium and hypertension control to numerous governmental and non-governmental organisations.

**Abbreviations:** CI, confidence interval; FOPL, front-of-package nutrition labelling; HW, health warning; NCD, non communicable disease; NFt, Nutrition Facts table; NMA, network meta-analysis; NS, Nutri-Score; NW, nutrient warning; OR, odds ratio; PICOS, population, intervention, comparator, outcome, and study design; RCT, randomised controlled trial; RMD, relative mean difference; SE, standard error; SIDE, separate indirect from direct evidence; TLS, traffic light labelling system; WHO, World Health Organization; WTP, willingness-to-pay.

−12.9%; energy per 100 g/ml: TLS, −3%; NS, −3.5%; NW, −3.8%), sodium (total sodium/salt: TLS, −6.4%; sodium/salt per 100 g/ml: NS: −7.8%), fat (total fat: NS, −15.7%; fat per 100 g/ml: TLS: −2.6%; NS: −3.2%), and total saturated fat (TLS, −12.9%; NS: −17.1%; NW: −16.3%) content of purchases. The impact of TLS, NS, and NW on purchasing behaviour could be explained by improved understanding of the nutrition information, which further elicits negative perception towards unhealthful products or positive attitudes towards healthful foods. Comparisons across label types suggested that colour-coded labels performed better in nudging consumers towards the purchase of more healthful products (NS versus NW: 1.51 [1.08, 2.11]), while warning labels have the advantage in discouraging unhealthful purchasing behaviour (NW versus TLS: 0.81 [0.67, 0.98]; HW versus TLS: 0.8 [0.63, 1]). Study limitations included high heterogeneity and inconsistency in the comparisons across different label types, limited number of real-world studies (95% were laboratory studies), and lack of long-term impact assessments.

## Conclusions

Our systematic review provided comprehensive evidence for the impact of colour-coded labels and warnings in nudging consumers' purchasing behaviour towards more healthful products and the underlying psychological mechanism of behavioural change. Each type of label had different attributes, which should be taken into consideration when making front-of-package nutrition labelling (FOPL) policies according to local contexts. Our study supported mandatory front-of-pack labelling policies in directing consumers' choice and encouraging the food industry to reformulate their products.

## Protocol registry

PROSPERO (CRD42020161877).

## Author summary

### Why was this study done?

- Interpretive front-of-package labelling is considered a cost-effective strategy to promote a more healthful diet and mitigate the burden of non communicable diseases (NCDs), and colour-coded labels and warning labels are the most adopted interpretive front-of-package labelling schemes worldwide.

- Prior to this study, evidence on the impact of each type of colour-coded labels and warning labels on modifying consumers purchasing behaviour was mixed.

- The feasibility and likely effectiveness of each label type applied in different contexts was unclear.

## What did the researchers do and find?

- This network meta-analysis summarised the currently available 118 peer-reviewed studies to update knowledge of the most mainstream interpretive front-of-package nutrition labelling (FOPL) schemes.

- We found that the traffic light labelling system (TLS), Nutri-Score (NS), nutrient warning (NW), and health warning (HW) were all able to direct consumers towards more healthful purchasing behaviour.

- Colour-coded labels (TLS and NS) performed better in promoting the purchase of more healthful products, while warning labels (NW and HW) had the advantage in discouraging unhealthful purchasing behaviour.

- The difference in consumers' behaviour could be explained by different underlying psychological mechanisms for each label.

## What do these findings mean?

- We provide more comprehensive evidence to guide policy-makers in choosing the optimal front-of-package labelling policies. This evidence synthesis may inform further generalisation of mandatory front-of-package labelling schemes and help to mitigate the burden of NCDs.

- Future studies should focus on the impact of FOPLs on dietary consumption in individuals, and industrial reformulation at the population level, especially in real-world settings and over a longer time frame. This will provide crucial, robust, and comprehensive evidence to guide policy making.

## Introduction

Suboptimal diets, linked to food environments that promote food and drink high in salt, sugar, and saturated fat, are a leading risk factor for death and disability worldwide, due to their relationship with non communicable diseases (NCDs) [1–6]. Nearly 8 million deaths in 2019 were attributable to dietary risk factors such as high salt intake and low wholegrain intake. To mitigate the healthcare burden resulting from NCDs, providing clear information about the nutritional profile of products is a recognised method to nudge consumers to more healthful food and drink options and exert pressure on manufacturers to carry out reformulation to improve the nutritional profile of their products [7,8]. As a minimum, many countries have mandatory nutrition tables on the back of food packaging [9], but the World Health Organization (WHO) additionally recommends front-of-pack nutrition labelling (FOPL) to promote healthful diets and help reduce NCD prevalence [10–13]. FOPL provides key nutritional information, typically including calorie, saturated fat, salt, and sugar content, in a visible format [14], and many countries have a voluntary FOPL system in place [15]. FOPL generally falls into 2 main categories—interpretive and noninterpretive. Interpretive labels present symbols, figures, or cautionary text to indicate the overall healthfulness or nutrient content of a product, such as the Nutri-Score (NS) label [16], Chilean style warning labels [17], Health Star Ratings (Australia and New Zealand), and the "traffic light" labelling system (TLS). For

example, the United Kingdom's TLS has red (high), amber (medium), or green (low) labels to indicate levels of total fat, saturated fat, sugars, and salt [18]. Noninterpretive FOPL systems, such as the Guideline Daily Amount, convey nutritional content as numbers rather than graphics, symbols, or colours, allowing consumers to create their own judgements on healthfulness.

At the time of completing this manuscript, a total of 31 countries had implemented interpretive FOPL systems, including 6 countries that had adopted mandatory warning labels on packaged foods and 3 countries that had utilised mandatory colour-coded FOPL systems [19]. Interpretive warning labels and colour-coded labels were the most adopted labels endorsed by governments. So far, real-world evidence is limited and mainly focused on the Chilean style warning label (a type of interpretative nutrient warning (NW) label). Observational studies found that Chile's Law of Food Labelling and Advertising, which included the mandatory implementation of warning labels nationwide, resulted in lower sales of beverages high in sugars, salt, saturated fat, or energy [20] and was likely to improve understanding and utilisation, especially in children [21]. However, no data are available on changes in consumption.

Based on the health communication theory and previous similar studies [22–26] (Fig 1), visual attention to labels is a prerequisite for the perception and understanding of FOPLs, but the mechanisms linking label interpretation and behavioural changes differ across types of FOPLs. Warning labels may elicit negative perception of unhealthful foods (e.g., perception of severe risk, lower grade of healthfulness, lower level or frequency of recommended consumption for unhealthful products) in the process of changing food choice. However, interpretative colour-coded labels (e.g., TLS) tend to modify purchasing behaviour by increasing the perception of healthfulness for healthier food options. The effects of FOPLs are also modified by study population demographics, knowledge of nutrition labels, frequency of grocery shopping, familiarity with the brand, level of weight consciousness and health status, product categories, and characteristics of labels [23–28]. For example, women and people with special diet needs or higher perceived nutrition knowledge are more likely to look at FOPLs; the provision of serving size information or percentage of recommended intake may add to consumers' difficulty in understanding FOPLs. Due to the abovementioned reasons, experimental evidence regarding the effectiveness of FOPL in modifying consumers' purchasing or consuming behaviour was mixed [14,23–40]. Meta-analyses showed that interpretative colour-coded FOPLs (e.g., TLS) significantly increased consumers' selection of more healthful options from a range of products, as well as decreased calorie and salt content of food purchased [31,41], but no synthesised results for warning labels were available. Experimental studies also provided some insights into the underlying mechanisms of how colour-coded and warning labels change consumers' behaviour, suggesting that warning labels effectively elicited negative emotions and raised health awareness, which led to the ultimate modifications in food choice; on the other hand, colour-coded labels (e.g., TLS) were indicated to have a favourable effect on increasing consumers' preference for healthier foods [25,27,28,31].

To summarise the existing findings and provide evidence for policy-makers in the proposed implementation or modification of food labelling policies, we aim to assess the impact of colour-coded and warning labels, the most studied and promising labels, on changes in both intended and actual purchasing and consumption behaviour. We also aim to gain insight into the underlying psychological mechanism based on the health communication theory to explore the heterogeneity across label types [22–26].

## Methods

The protocol of this systematic review was registered on PROSPERO (CRD42020161877).

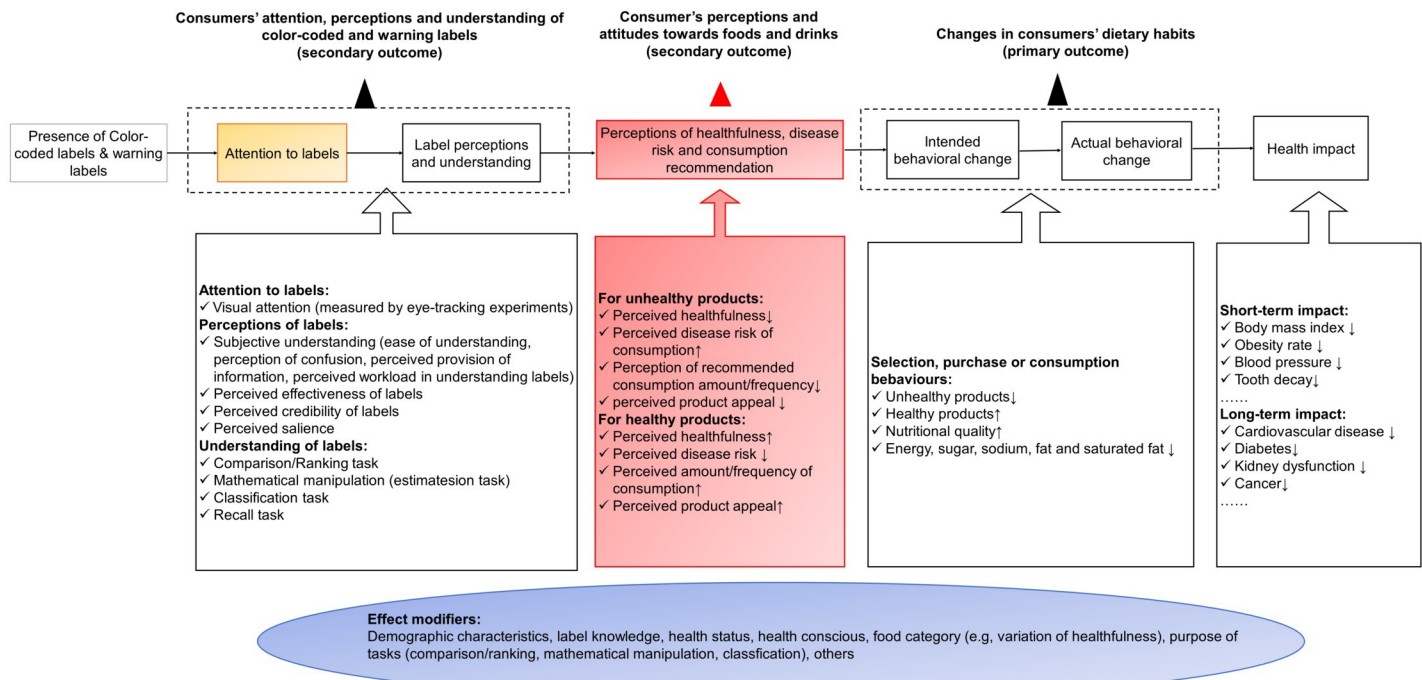

**Fig 1. The logic model of the impact of colour-coded labels and warning labels on consumers' behaviour based on the health communication theory.** Attention to labels is a prerequisite for understanding of labels and forming perception towards labels (amber box), but the mechanisms connecting changes in label understanding and perception with behavioural changes differ across FOPL types. Warning labels (e.g., NW and HW) elicit negative perception of food products (e.g., perception of severe risk, negative emotion) in motivating changes in food choice, while interpretive colour-coded labels (e.g., TLS and NS) improve the perception of healthfulness for healthier food options (red box). Demographic characteristics, knowledge of labels, nutrition and health, food categories, and experiment settings also modify the effects of FOPLs throughout the mechanism (blue box). FOPL, front-of-package nutrition labelling; HW, health warning; NS, Nutri-Score; NW, nutrient warning; TLS, traffic light labelling system.

## Search strategy

The systematic review was conducted and reported in compliance with the Preferred Reporting Items for Systematic Reviews and Meta-analyses (PRISMA) Network Meta-analysis guideline (S1 PRISMA Checklist). We searched 4 databases (PubMed, Embase via Ovid, Cochrane Central Register of Controlled Trials, and SCOPUS) initially on 26 November 2019 using the following search strategy: ("food and beverages" OR "food" OR "drink" OR "beverage" OR "meal" OR "nutrition" OR "menu" OR "restaurant") AND ("warning" OR "traffic light" OR "Wheel of health" OR "colored GDA" OR "coloured GDA" OR "5 CNL" OR "Color Nutrition" OR "Nutriscore" OR "Nutri-score" OR "simplified nutrition labelling system" OR "SENS" OR "colour-coded" OR "color-coded" OR "colour coding" OR "color coding" OR "red label" OR "green label" OR "Evolved Nutrition Label"). There was no restriction on language. The search was updated on 24 May 2021 to capture the latest publications up to the submission of manuscript. Additional articles were identified by reviewing the bibliographic reference of identified articles. Articles of all types published from 1 January 1990 were included at this stage.

## Inclusion and exclusion criteria, data extraction, and bias assessment

Original studies were selected based on the population, intervention, comparator, outcome, and study design (PICOS) framework (Table 1). All randomised controlled trials (RCTs) and quasi-experimental studies assessing 4 types of interpretative FOPLs (TLS, NS, NW, and HW) were included. In addition, cross-sectional and cohort studies were included in the assessment

**Table 1. PICOS criteria for inclusion and exclusion of studies.**

| Criteria | Description |
|---|---|
| **Population** | General population |
| **Intervention** | **Inclusion:** |
| | 1. All colour-coded labels and warning front-of-package labels: TLS, NS, NW, and HW; |
| | 2. For each of TLS, NW, and HW, different formats were also separately included into analysis: summary TLS versus nutrient-specific TLS, negative message framing versus positive message framing, and textual warning versus textual + nontextual warning (e.g., picture of tooth decay, stop sign). |
| | **Exclusion:** |
| | 1. Other colour-coded labels (e.g., Wheel of health, ENL, and SENS) were not included because they had been scrapped or no longer actively developed due to political reasons or lack of public popularity. |
| | 2. Studies that combined effects of nutrition labels of included types and excluded types (e.g., TLS + HSR). |
| | 3. Studies investigated the combined effect of nutrition labelling and other intervention (e.g., TLS + food placement). |
| | 4. Studies that assessed only the effect of nutritional education using traffic light label as educational tools were not included. |
| | 5. Studies that applied colour-coding schemes (traffic light label score, NS) only to categorise the healthfulness of products were excluded. |
| **Comparator** | 1. Absence of nutritional information (no-label) |
| | 2. Only NFt present |
| **Outcome** | All measured outcomes were included based on the health communication model. |
| **Study design** | 1. RCTs, including randomised controlled crossover trials and clustered randomised trials |
| | 2. Quasi-experimental studies: |
| | • Nonequivalent control group design |
| | • Interrupted time series design |
| | • One-group prepost design |
| | • Choice-based conjoint studies |
| | 3. Cross-sectional studies evaluating at least 2 intervention labels, or 1 intervention labels and control condition (no-label exposure or NFt): only eligible for the secondary outcomes. |

ENL, evolved nutrition label; FOPL, front-of-package nutrition labelling; HSR, Health Star Rating; HW, health warning; NFt, Nutrition Facts table; NS, Nutri-Score; NW, nutrient warning; PICOS, population, intervention, comparator, outcome, and study design; RCT, randomised controlled trial; TLS, traffic light labelling system.

of secondary outcomes (see "Measurement of outcomes"). We followed the criteria of a previous meta-analysis, which set a control group by merging both Nutrition Facts table (NFt, also known as Nutrition Information Panel in some countries) and no-label condition into the control group in the main analyses, so as to increase the statistical power and precision of network meta-analysis (NMA) [25,42]. Studies that featured other interpretive or noninterpretive FOPLs (e.g., Guideline Daily Amount, Health Star Rating) as the reference group were not eligible as they differed too much from the control group specified (i.e., back-of-package labels and no-label control). Reviews, study protocols, and conference abstracts were also excluded.

To select eligible studies, titles, abstracts, and main texts were reviewed based on the eligibility criteria. The bibliographic references of the eligible articles were also reviewed to identify additional articles missed out by the database search strategy. Data extraction was carried out using an extraction spreadsheet, consisting of the following variables: publication year, language, country of study, study design, sample size, setting (real-world setting that utilises the sales data before and after the implementation of the actual labelling intervention in retail

outlets, or controlled laboratory settings based on a virtual food environment), race, mean age or age range, percentage of female, education, income, occupation, socioeconomics, investigated food types, number of food types (single or multiple), intervention condition, control condition, access to NFt for both intervention and control groups, time pressure, outcome, measure of outcome, and effect estimate.

Bias was assessed using the revised Cochrane risk-of-bias tool for randomised trials (ROB 2) for RCTs [43], Risk Of Bias In Non-randomised Studies of Interventions (ROBINS-I) for quasi-experimental studies [44], and National Heart, Lung, and Blood Institute (NHLBI) Quality Assessment Tool for Observational Cohort and Cross-Sectional Studies [45] for cross-sectional studies. The overall risk-of-bias judgement for each study was summarised as low risk of bias if the study received the assessment of "low risk" for all domains, and high risk of bias if the study was judged to be at high risk in at least one domain. Studies with insufficient information to assess the risk of bias for some domains was categorised into "some concerns" (S1 Text and S10–S12 Tables).

Inclusion and exclusion, data extraction, and risk of bias were first assessed by 2 independent reviewers (JS and MB). A total of 16 discrepancies (13.6%) was found and were referred to a third reviewer (MT).

## Measurement of outcomes

Based on the theory conceptualised in Fig 1, we grouped the outcome measures into 3 categories: (1) changes in consumers' purchasing and consumption behaviour; (2) consumers' perception and attitudes towards products; and (3) consumers' attention, understanding, and perception of colour-coded and warning labels (S1 Table). The primary outcomes of our systematic review were measures regarding the changes in consumers' purchasing and consuming behaviour, which contain the probability of choosing less healthful or more healthful products, self-reported ratings of purchase intention, overall healthfulness of products purchased, and energy and nutrient (salt/sodium, sugar, fat, and saturated fat) content of products purchased/consumed. The healthful and unhealthful products were grouped based on the levels of salt/sodium, sugar, saturated fat, and calories indicated by the front-of-package labelling systems. Products with warning texts, symbols, or colours (e.g., red in multiple traffic light label) indicating high levels of sugar, salt, saturated fat, or calorie content per 100 g or per 100 ml (per 100 g/ml) relative to reference intake, or overall low healthfulness (e.g. red in single traffic light, or orange and red [D and E] in NS), were defined as unhealthful. A list of measures was included in the other 2 secondary outcome domains (S1 Table). Data on the comparisons between intervention labels and control group or pairwise comparisons between any 2 types of intervention labels (e.g., TLS versus NW) were collected. Depending on the results reported in the original studies, we calculated odds ratios (ORs) and 95% confidence intervals (95% CIs) as the summary estimates for categorical outcomes, and relative mean differences (RMDs, the percentage of change comparing intervention and control group) plus standard errors (SEs) for continuous outcomes [46] (S1 Text and S1 Table).

## Data synthesis and network meta-analysis

As multiple nutrition labels were evaluated, and several multiarmed trials were included in our analysis, we used the frequentist NMA method to synthesise studies and make both direct (observed) and indirect (unobserved) comparisons of multiple interventions [47,48]. Random-effect models were fitted in the NMA as we assumed the heterogeneity in our network model was high. Cochran's $Q$-statistic and Higgin's and Thompson's $I^2$ were applied to assess the degree of total heterogeneity in the network model, which was further divided into within-

(conventional between-study heterogeneity) and between-design (overall inconsistency) variations, respectively. To test the transitivity and consistency assumptions underlying NMA, we further calculated the Q-statistic in a full design-by-treatment intervention random-effect inconsistency model [49]. When there was inconsistency between the effect estimate of direct and indirect comparisons, we took the direct effect estimate into consideration instead. We also applied the method of separate indirect from direct evidence (SIDE), generating the proportion of direct evidence for each comparison (local inconsistency) [50]. A detailed explanation of the NMA can be found in the S1 Text.

In addition to the analysis of 4 main label types (TLS, NS, NW, and HW) for each outcome measure, we also synthesised the evidence grouped by labelling formats and framing. TLS was further divided into summary TLS and detailed TLS (summary TLS was defined as the single TLS indicating the overall nutritional quality as good, medium, and poor using predefined algorithms; detailed TLS included information on individual nutrients). NW and HW were further categorised into textual or nontextual NW/HW (nontextual NW/HW included graphs and symbols that alert consumers to the high levels of nutrient content or health risks associated with high nutrient content). In addition, HW was also divided into negative or positive, depending on the framing of the warning (positive framing highlighted the health benefits resulting from lowered consumption of a nutrient; negative emphasised the harm associated with excessive consumption of a nutrient). To explore the presence of effect modifiers leading to overall inconsistency between direct and indirect comparisons, subgroup analyses by sex composition (primarily female: female >60%; primary male: female <40%; otherwise equally distributed), age group (primary adults: >70% aged 18 or older; primary children and adolescent: >70% aged less than 18; otherwise general population), setting (real-world or laboratory setting), types of products investigated (single/multiple), and display NFt both in intervention were conducted if at least 2 studies were available for quantitative synthesis in every stratum for each primary outcome measure. Sensitivity analyses of the primary outcomes were conducted by (1) evaluating only RCTs and (2) removing studies using NFt control. We did not perform meta-regression adjusting the effect modifier variables, given that there were a low number of studies available for most outcomes.

A comparison-adjusted funnel was plotted to assess the risk of publication bias under the priori hypothesis that studies identifying the superiority of newly developed labels to an existing labelling system tend to be published in higher frequency [50]. The funnel plot was only created when there were at least 10 studies available for each outcome, as recommended by Cochrane [51]. The results were considered statistically significant when $p < 0.05$.

All the statistical analyses were implemented in R v3.5.1. The NMA was conducted using the *netmeta v1.2–1* R package [48].

## Results

### Study characteristics

Using the search strategy mentioned above, we obtained 15,058 records from the 4 databases. After removing duplicate articles and excluding ineligible articles based on the title, abstract, and main text, and identifying studies in the bibliographic references of the eligible articles, 156 studies nested in 138 articles remained (S2 Table), of which 22 studies were excluded because the results reported in original papers were incomplete (e.g., missing standard deviations/SEs) and the authors were unable to provide the complete results after contacting via email. In total, 134 studies were eligible for quantitative synthesis (Fig 2). The details of included studies are presented in S1 Data.

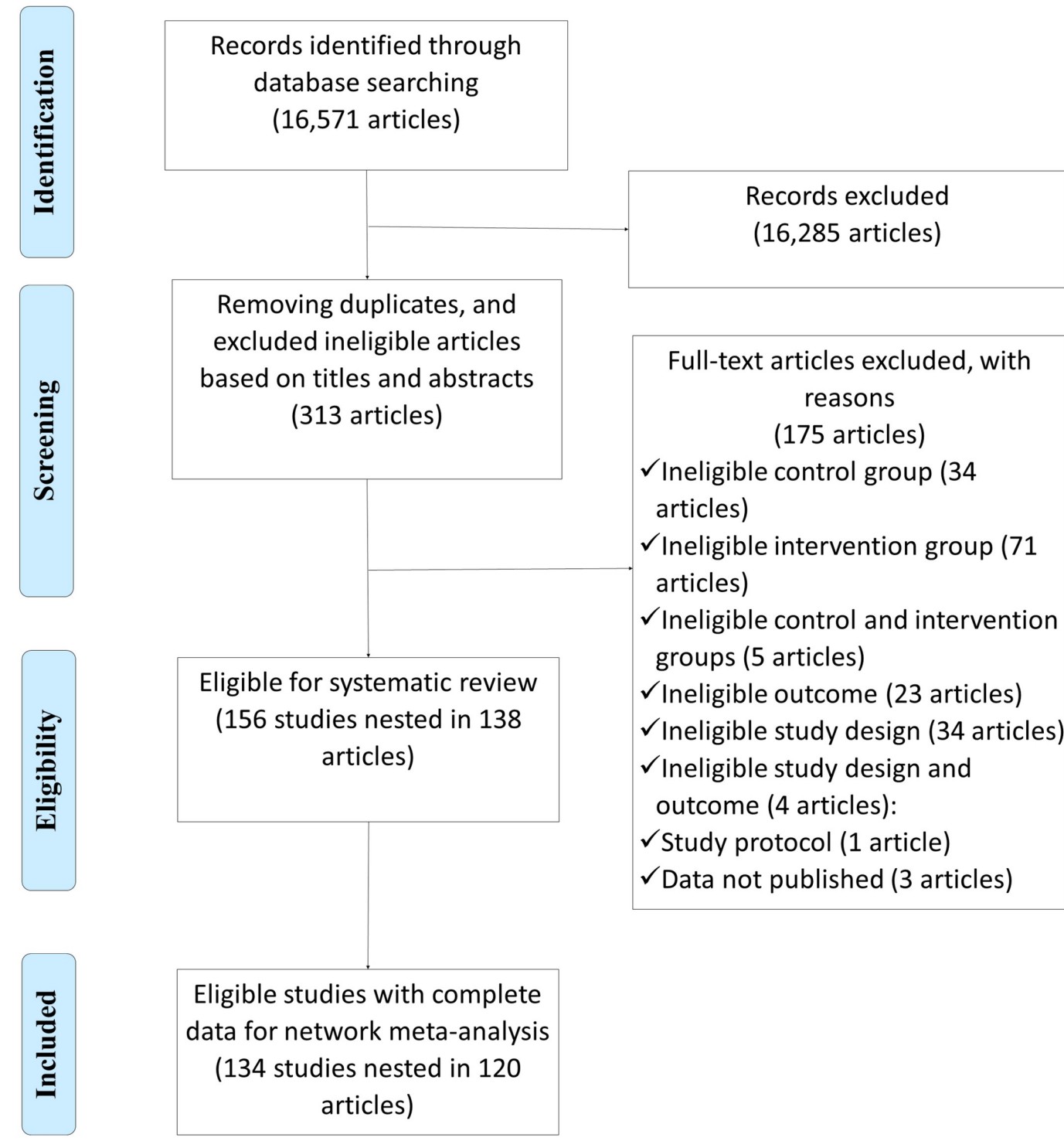

**Fig 2. Flow diagram of literature searching and screening.**

Of the 156 studies eligible for systematic review, 154 were published in English, and 2 in Spanish. The majority of articles were published 2018 to present (64%), and the most common study populations were in the regions of Europe (31%), Latin America (28%), and North

America (24%). Most studies were carried out in a laboratory setting (95%). Of the studies based in real-world settings (5%), half were conducted in the out-of-home sector and half in retail outlets. Most studies were RCTs (65%). None of the studies were industry funded.

The majority of studies assessed TLS (62%), while 40% studies investigated NW, 22% studied NS, and 17% evaluated HW (Fig 3). For comparators, 80% studies used a no-label condition. Most studies (78%) did not provide NFt along with the labels in the intervention group. Most studies used multiple types of foods and drinks to assess the effect of labels (56%), and 40% focused on a single type. Nearly all studies leveraged individual-level data (97%), and only 4 studies were based on sales data. Most studies had a population of primarily adults (88%) (Table 2).

The comparison-adjusted funnel showed that there was no publication bias detected for most outcomes (S4–S6 Figs). For a few outcomes (e.g., probability of selecting more healthful options, energy and sugar of purchased products, objective understanding and perceived effectiveness of FOPLs) presenting funnel asymmetry, we cannot simply predict them as publication biases either as controversy remains in the test accuracy as the appearance of the plot may be affected by the coding of outcomes and choice of measures [52].

## Consumers' attention, perception, and understanding of colour-coded and warning labels

**Objective understanding.** The objective understanding of labels was measured by 4 indexes that represent different aspects of label interpretation: (1) comparison or ranking; (2) recall; (3) classification; and (4) mathematical manipulation (estimation) of overall healthfulness and nutrient content. TLS was the only label observed to improve understanding of nutrition information in all 4 types of tasks (Tables 3 and S5 and S2 Fig). NS was also observed to boost the participants' ability to compare/rank products and estimate overall healthfulness, and NW was associated with improved ability to compare/rank and classify overall healthfulness. When colour-coded labels and warning labels were compared against each other, TLS was found to outperform NW in classifying sugar and saturated fat than NW, and NS was linked to better capacity in the comparison/ranking of overall healthfulness than NW. Comparisons between the 2 colour-coded labels suggested that NS might perform better than TLS in comparing/ranking overall healthfulness task.

**Subjective understanding.** We assessed different dimensions of subjective understanding of the labels in terms of ease of understanding (16 studies), elimination of confusion (5 studies), perceived provision of information (13 studies), and perceived workload (5 studies) in label processing (S5 Table and S3 Fig). TLS caused more confusion for consumers (OR and 95% CI for elimination of confusion: 0.66 [0.45, 0.97]) but was seen as providing sufficient information (3.08 [1.6, 5.93]). NS was considered easy to understand (1.84 [1.19, 2.85]) and providing sufficient information for consumers (2.44 [1.03, 5.78]). NW was evaluated as the easiest to understand (NW versus control: 2.82 [1.82, 4.36]; NW versus TLS: 2.03 [1.32, 3.12]).

**Perceived effectiveness and credibility.** Thirteen studies were included for the evaluation of perceived effectiveness, while 8 studies for the credibility assessment (S5 Table and S3 Fig). TLS, NS, and HW were all perceived effective by consumer surveys (OR and 95% CI: TLS, 3.2 [1.43, 7.19]; NS, 3.92 [1.42, 10.8]; HW: 4.75 [1.03, 21.88]). Comparing colour-coded labels against the warning labels suggested that NS was believed to be more effective than NW. None of the FOPLs were thought more credible than the control group.

**Salience and visual attention.** Several self-reported measures were applied to evaluate the salience of labels, including how conspicuous labels were (10 studies included using self-reported ratings of the question "to what degree to you think the label stands out") and how

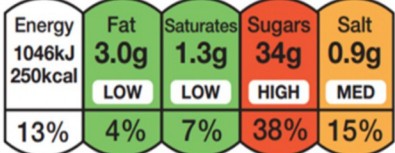

**A. Detailed Traffic Light label**

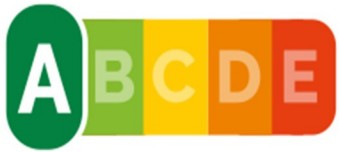

**B. Nutri-score**

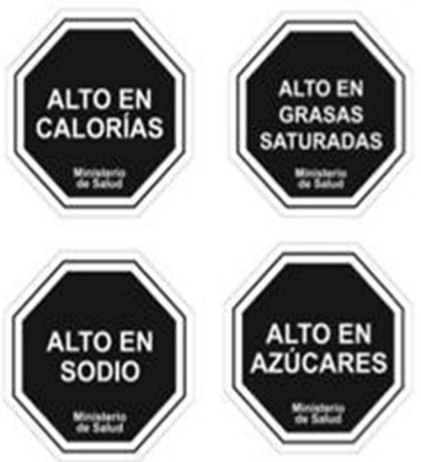

**C. Chilean warning labels system**

STATE OF CALIFORNIA SAFETY WARNING:
Drinking beverages with added sugar(s)
contributes to obesity, diabetes, and tooth decay.

**D. California safety warning**

**Fig 3.** The design of colour-coded labels (A, B) and warning labels (C, D). (A) UK traffic light label uses green, amber, and red to represent low, moderate, and high levels of fat, saturated fat, sugar, and sodium per 100 g or per 100 ml on the front of food packages, respectively, with addition of the % reference intake value for calorie and each nutrient. (B) NS includes a colour spectrum ranging from dark green to dark orange with letters from A to E. Products assigned an "A" were considered to have the best nutritional quality while "E" the poorest. (C) Chilean warning labels is a type of NW with a textual warning "high in [calorie/nutrient]" presented on the octagonal signs in a black-and-white design. (D) California safety warning is a type of HWs designed for sugar-sweetened beverages with ≥75 calories contributed by added sugar. HW, health warning; NS, Nutri-Score; NW, nutrient warning.

frequently they were noticed (5 studies assessed the frequency of participants claiming that they noticed the labels) and recalled (3 studies evaluated if participants correctly recalled the labels). TLS, NW, NW, and HW were all noticed more frequently than the control group (OR and 95% CI: TLS, 2.58 [1.65, 4.02]; NS, 5.65 [2.84, 11.22]; NW, 3.04 [1.87, 4.95]; HW, 6.09 [3, 12.36]). Only NS was perceived to be conspicuous (7.53 [1.78, 31.89]) and was more likely to be recalled (3.46 [2.74, 4.37]) than the control. Comparing colour-coded labels and warning labels showed that NS and HW was noticed more often than other options. NS was correctly recalled more frequently than TLS (S5 Table and S3 Fig).

To measure visual attention more accurately and precisely, some studies utilised eye-tracking devices to record the eye movement, and a fixation was defined as low velocity of eye movement (S5 Table and S3 Fig). Only TLS and NS were evaluated in 5 eye-tracking studies. Our analysis indicated that TLS attracted fixations more frequently and quickly with longer duration. The use of TLS also delayed the fixation on NFt and reduced the visual attention on NFt. According to our findings, TLS was perceived to provide more nutrition information and thus required more time to interpret. In addition, the dependence on conventional NFt was

**Table 2. Characteristics of studies included in the systematic review (*n* = 135).**

| Study characteristics | Number of studies | Proportion (%) |
|---|---|---|
| **Publication year** | | |
| 2021 | 17 | 10.9 |
| 2020 | 26 | 16.67 |
| 2019 | 32 | 20.51 |
| 2018 | 25 | 16.03 |
| 2017 | 8 | 5.13 |
| 2016 | 9 | 5.77 |
| 2015 | 9 | 5.77 |
| 2014 | 8 | 5.13 |
| 2013 | 5 | 3.21 |
| 2012 | 4 | 2.56 |
| 2011 | 5 | 3.21 |
| 2010 and before | 8 | 5.13 |
| **Language** | | |
| English | 154 | 98.72 |
| | 2 | 1.28 |
| **Country** | | |
| US | 27 | 17.31 |
| Uruguay | 19 | 12.18 |
| France | 16 | 10.26 |
| Canada | 11 | 7.05 |
| Australia | 10 | 6.41 |
| UK | 9 | 5.77 |
| New Zealand | 7 | 4.49 |
| Chile | 7 | 4.49 |
| Brazil | 7 | 4.49 |
| Germany | 6 | 3.85 |
| Ecuador | 5 | 3.21 |
| Switzerland | 3 | 1.92 |
| Mexico | 3 | 1.92 |
| Other | 19 | 16.67 |
| **Region** | | |
| Europe | 49 | 31.41 |
| Latin America | 43 | 27.56 |
| North America | 38 | 24.36 |
| Oceania | 17 | 10.9 |
| Asia | 4 | 2.56 |
| Africa | 1 | 0.64 |
| Multiple | 4 | 2.56 |
| **Setting** | | |
| Laboratory | 148 | 94.87 |
| Real-world | 8 | 5.13 |
| Out-of-home sectors* | 4 | 2.56 |
| Retailer outlets | 4 | 2.56 |
| **Study design** | | |
| RCT | 101 | 64.74 |
| Quasi-experimental studies and cross-sectional survey | 55 | 35.26 |

(*Continued*)

**Table 2.** (Continued)

| Study characteristics | Number of studies | Proportion (%) |
|---|---|---|
| **Intervention label** | | |
| TLS | 97 | 62.18 |
| NW | 62 | 39.74 |
| NS | 35 | 22.44 |
| HW | 27 | 17.31 |
| **Control condition** | | |
| No label | 125 | 80.13 |
| NFt | 12 | 7.69 |
| Compared against each other colour-coded or warning labels | 19 | 12.18 |
| **Types of foods and drinks tested** | | |
| Multiple | 87 | 55.77 |
| Single | 62 | 39.74 |
| Not stated | 7 | 17.31 |
| **Research data** | | |
| Individual data | 152 | 97.44 |
| Sales data | 4 | 2.56 |
| **Age group** | | |
| Primarily adults | 134 | 88.16 |
| Primarily children or adolescents | 5 | 3.29 |
| Mixed population made up of adults, children, and adolescents | 5 | 3.29 |
| Not stated | 8 | 5.26 |
| **Sex** | | |
| Mixed | 85 | 55.92 |
| Mostly female | 60 | 39.47 |
| Mostly male | 1 | 0.66 |
| Not stated | 6 | 3.95 |
| **Education[#]** | | |
| High | 47 | 30.92 |
| Low | 45 | 29.61 |
| Not stated | 60 | 39.47 |
| **Individual/Familial income** | | |
| High | 1 | 0.66 |
| Low | 6 | 3.95 |
| Mixed | 48 | 31.58 |
| Not stated | 97 | 63.82 |
| **Occupation** | | |
| Undergraduate students | 6 | 3.95 |
| Primary school students | 1 | 0.66 |
| Mixed | 11 | 7.24 |
| Not stated | 134 | 88.16 |
| **Presence of NFt along with intervention label** | | |
| Yes | 34 | 21.79 |
| No | 122 | 78.21 |
| **Risk of bias** | | |
| High | 107 | 68.59 |
| Moderate/Some concerns | 35 | 22.44 |

(*Continued*)

**Table 2.** (Continued)

| Study characteristics | Number of studies | Proportion (%) |
|---|---|---|
| Low | 14 | 8.97 |

*Out-of-home sector includes any outlet where food or drink is prepared for immediate consumption by consumers, such as restaurants, cafes, and takeaways.

#Education was classified as "high" if >50% of study population completed university or college education, otherwise classified into "low."

HW, health warning; NFt, Nutrition Facts table; NS, nutri-score; NW, nutrient warning; RCT, randomised controlled trial; TLS, traffic light labelling system.

lessened due to the salience and perceived provision of information of TLS. NS also captured visual attentions faster than the control condition, but total duration and number of fixations on the label were also reduced compared with NFt control, probably due to the nature of summarised label simplifying the interpretation of nutrition information.

## Consumers' perception and attitudes towards foods and drinks

To explore the mechanisms underpinning behavioural changes (formation of attitudes), we further assessed the influence of different colour-coded and warning labels on consumers' perception and attitudes towards products, based on perceived healthfulness and risks, perceived recommended amount and frequency of consumption, self-reported product appeal, and willingness-to-pay (WTP).

The results indicated that TLS, NW, and HW all reduced the perception of healthfulness for less healthful products (RMD and 95% CI: TLS, −0.077 [−0.116, −0.038]; NW, −0.224 [−0.263, −0.186]; HW, −0.126 [−0.16, −0.092]) and reduced the perceived recommended amount to consume unhealthful foods (perceived recommended amount to consume: TLS, −0.05 [−0.054, −0.046]; NS, −0.08 [−0.082, −0.078]; NW, −0.47 [−0.509, −0.431]), but NW performed significantly better than TLS and HW (perceived healthfulness for unhealthful products: NW versus TLS, −0.147 [−0.195, −0.1]; NW versus NS, −0.188 [−0.301, −0.075]; perceived recommended amount of unhealthful products to consume: NW versus TLS, −0.42 [−0.459, −0.381]; NW versus NS, −0.39 [−0.429, −0.351]; perceived recommended frequency of unhealthful products to consume: NW versus TLS, −0.41 [−0.449, −0.371]; NW versus NS, −0.455 [−0.529, −0.381]) (Tables 4 and S4 and S1 Fig). NW and HW also increased the perceived disease risk of unhealthful products, as well as reduced the appeal of unhealthful products (Tables 5 and S4). On the other hand, however, TLS and NS promoted the perceived healthfulness for more healthful products better than NW (NW versus TLS, −0.145 [−0.282, −0.008]; NW versus NS, 0.201 [−0.399, −0.003]) (Tables 4 and 5 and S4 and S1 Fig).

WTP is an economic concept used to evaluate consumers' demand for a product [53]. In our systematic review, 3 studies considered WTP of less healthful products, suggesting that NS significantly reduced the WTP for less healthful products by 16%, but no significant effect was observed for textual NW or HW. Another 3 studies investigated the WTP for more healthful products but did not find evidence for change in WTP when labelled with TLS, textual NW, or HW (S1 Data).

## Changes in consumers' purchasing and consuming behaviour or intentions

The NMA revealed that all colour-coded and warning labels were significantly associated with changes in purchasing behaviour. Warning labels also have a significant effect on purchasing intention (Fig 4 and S3 Table). NS and NW were both associated with an increasing overall

**Table 3. The network effects of objective understanding of different label types.**

| Outcome | Comparison | Number of comparisons | Direct estimate (OR and 95% CI) | Indirect estimate (OR and 95% CI) | Network estimate (OR and 95% CI) | Direct evidence proportion |
|---|---|---|---|---|---|---|
| | | | **Comparison/Ranking of overall healthfulness and nutrient content** | | | |
| Overall healthfulness | NS vs. control | 14 | 4.84 (3.36, 6.98) | 3.52 (1.72, 7.19) | **4.53 (3.28, 6.28)** | 0.79 |
| | NS vs. NW | 7 | 1.62 (0.97, 2.72) | 1.93 (1.08, 3.44) | **1.75 (1.19, 2.57)** | 0.56 |
| | NS vs. TLS | 8 | 1.38 (0.85, 2.24) | 1.87 (1.07, 3.26) | **1.57 (1.09, 2.27)** | 0.57 |
| | NW vs. control | 14 | 2.22 (1.54, 3.2) | 3.92 (2.15, 7.17) | **2.59 (1.89, 3.54)** | 0.73 |
| | NW vs. TLS | 10 | 0.91 (0.59, 1.41) | 0.88 (0.51, 1.53) | 0.9 (0.64, 1.26) | 0.61 |
| | TLS vs. control | 21 | 2.91 (2.15, 3.94) | 2.75 (1.44, 5.24) | **2.88 (2.19, 3.79)** | 0.82 |
| Energy | TLS vs. control | 2 | 3.1 (1.36, 7.08) | . | **3.1 (1.36, 7.08)** | 1.00 |
| Sodium/Salt | TLS vs. control | 2 | 2.92 (1.06, 8.06) | . | **2.92 (1.06, 8.06)** | 1.00 |
| | | | **Mathematical manipulation (estimation) of overall healthfulness and nutrient content** | | | |
| Overall healthfulness | NS vs. control | 0 | . | 2.57 (1.42, 4.65) | **2.57 (1.42, 4.65)** | 0 |
| | NS vs. TLS | 1 | 0.88 (0.69, 1.14) | . | 0.88 (0.69, 1.14) | 1.00 |
| | TLS vs. control | 1 | 2.91 (1.7, 4.98) | . | **2.91 (1.7, 4.98)** | 1.00 |
| Energy | TLS vs. control | 2 | 2.4 (0.39, 14.69) | . | 2.4 (0.39, 14.69) | 1.00 |
| Fat | TLS vs. control | 1 | 1.32 (0.87, 2) | . | 1.32 (0.87, 2) | 1.00 |
| | | | **Recall of overall healthfulness and nutrient content** | | | |
| Overall healthfulness | TLS vs. control | 1 | 1.34 (0.8, 2.22) | . | 1.34 (0.8, 2.22) | 1.00 |
| Energy | TLS vs. control | 1 | 1.55 (1.14, 2.11) | . | **1.55 (1.14, 2.11)** | 1.00 |
| | | | **Classification of overall healthfulness and nutrient content** | | | |
| Overall healthfulness | NW vs. control | 4 | 2.63 (1.42, 4.88) | 6.46 (1.72, 24.18) | **3.09 (1.77, 5.41)** | 0.82 |
| | NW vs. TLS | 2 | 1.81 (0.75, 4.39) | 1.02 (0.4, 2.63) | 1.39 (0.73, 2.65) | 0.53 |
| | TLS vs. control | 5 | 2.45 (1.4, 4.27) | 1.25 (0.31, 4.98) | **2.23 (1.33, 3.73)** | 0.86 |
| Energy | TLS vs. control | 3 | 4.53 (0.63, 32.37) | . | 4.53 (0.63, 32.37) | 1.00 |
| Sodium/Salt | TLS vs. control | 4 | 3.97 (1.89, 8.32) | . | **3.97 (1.89, 8.32)** | 1.00 |
| Sugar | NW vs. control | 1 | 1.31 (0.56, 3.06) | . | 1.31 (0.56, 3.06) | 1.00 |
| | NW vs. TLS | 0 | . | 0.31 (0.11, 0.89) | **0.31 (0.11, 0.89)** | 0 |
| | TLS vs. control | 2 | 4.28 (2.23, 8.18) | . | **4.28 (2.23, 8.18)** | 1.00 |
| Fat | TLS vs. control | 3 | 3.09 (1.26, 7.6) | . | **3.09 (1.26, 7.6)** | 1.00 |
| Saturated fat | NW vs. control | 1 | 1.8 (0.96, 3.38) | . | 1.8 (0.96, 3.38) | 1.00 |
| | NW vs. TLS | 0 | . | 0.4 (0.18, 0.88) | **0.4 (0.18, 0.88)** | 0.00 |
| | TLS vs. control | 2 | 4.54 (2.78, 7.43) | . | **4.54 (2.78, 7.43)** | 1.00 |

CI, confidence interval; HW, health warning; NS, Nutri-Score; NW, nutrient warning; OR, odds ratio; TLS, traffic light labelling system.

healthfulness of products purchased. TLS, NS, and NW all were associated with purchasing lower energy, sodium/salt, total fat, or saturated fat (Figs 5 and 6 and S3 Table). Only one study looked at the outcome of consumption (S3 Table), thus we did not include it in our meta-analysis. This study suggested that TLS was not significantly associated with change in energy consumption.

Eight studies conducted in real-world settings examined the association between FOPLs and sales of unhealthful or healthful products (S7 Table). NW was linked to reduced probability of purchasing/selecting of unhealthful items (OR and 95% CI: 0.50 [0.34, 0.73]), while TLS was associated with increased likelihood of purchasing/selecting more healthful products (1.32 [1.02, 1.72]).

When colour-coded labels and warning labels were compared against each other, we found that NW appeared to outperform TLS in discouraging purchasing unhealthful products (OR and 95% CI: 0.81 [0.67, 0.98]), reducing intended purchase of unhealthful products (RMD and 95% CI: −0.197 [−0.352, −0.042]), and lowering total amount of energy purchased (RMD and 95% CI: −0.064 [−0.125, −0.004]). NW was also observed to perform better than NS in improving overall healthfulness (RMD and 95% CI: 0.127 [0.029, 0.225]) and reduce total energy (RMD and 95% CI: −0.07 [−0.008, −0.131]) and saturated fat (RMD and 95% CI: −0.156 [−0.264, −0.049]) in shopping basket, but NS appeared to encourage the purchase of healthful products better than NW (OR and 95% CI: 1.51 [1.08, 2.11]) (S3 Table).

Comparisons within colour-coded labels or warning labels suggested moderate difference as well. NW reduced the purchase intention for unhealthful products more than HW (RMD and 95% CI: 0.178 [0.002, 0.355]), but HW performed better in boosting the intended purchase of healthful products (0.429 [0.058, 0.8]). TLS did not differ from NS much in terms of the behavioural changes. When comparing the results of different subtypes of TLS (summary versus nutrient-specific TLS), NW (textual NW versus nontextual NW), and HW (textual HW versus nontextual HW, negative message versus positive message), we did not find any difference on the behavioural changes.

Considering the evident heterogeneity and inconsistency for some outcome measures that might violate the consistency and transitivity assumptions required for an NMA (S3 Table), our findings should be interpreted with caution for these outcomes.

## Discussion

This systematic review evaluated the impact of colour-coded and warning labels on consumers' purchasing behaviour, the psychological mechanism underpinning purchasing modification, including consumers' perception and liking for foods and drinks, as well as the understanding and evaluation of label attributes. We found that all colour-coded and warning labels appeared to have beneficial effects by encouraging the purchase of more healthful products, reducing the purchase of less healthful options, improving overall nutritional quality, and reducing the energy, sodium/salt, fat, and saturated fat content of processed foods and drinks purchased/chosen. Based on the health communication theory, the results suggested that colour-coded and warning labels successfully drew more of consumers' attention than the control condition and improved consumers' understanding of nutrition information. The labels also modified perceived healthfulness, recommended consumption amount, and frequency of consumption for products. These mechanisms can establish more healthful purchasing behaviour by improving both the nutritional quality and nutrient content purchased/chosen by consumers.

Despite the heterogeneity in label types, labelling formats, position, study population, study design, and experimental settings across studies [14,23–40], FOPLs were generally considered to have positive effects on guiding consumers in making more healthful food choices,

**Table 4. The network effects of different label types on the perceived healthfulness of products.**

| Outcome | Comparison | Number of comparisons | Direct estimate (RMD and 95% CI) | Indirect estimate (RMD and 95% CI) | Network estimate (RMD and 95% CI) | Proportion of direct evidence |
|---|---|---|---|---|---|---|
| Less healthful products | HW vs. control | 10 | −0.125 (−0.159, −0.091) | −0.289 (−0.712, 0.133) | **−0.126 (−0.16, −0.092)** | 1.00 |
| | HW vs. NS | 0 | . | −0.089 (−0.204, 0.025) | −0.089 (−0.204, 0.025) | 0 |
| | HW vs. NW | 1 | 0.099 (−0.061, 0.259) | 0.098 (0.045, 0.152) | **0.098 (0.048, 0.149)** | 0 |
| | HW vs. TLS | 0 | . | −0.049 (−0.1, 0.003) | −0.049 (−0.1, 0.003) | 0 |
| | NS vs. control | 1 | −0.078 (−0.383, 0.227) | −0.031 (−0.148, 0.087) | −0.037 (−0.146, 0.073) | 0.05 |
| | NS vs. NW | 0 | . | 0.188 (0.075, 0.301) | **0.188 (0.075, 0.301)** | 0 |
| | NS vs. TLS | 1 | 0.046 (−0.064, 0.156) | −0.001 (−0.309, 0.306) | 0.041 (−0.063, 0.144) | 0.95 |
| | NW vs. control | 6 | −0.215 (−0.255, −0.175) | −0.327 (−0.46, −0.194) | **−0.224 (−0.263, −0.186)** | 0.99 |
| | NW vs. TLS | 3 | −0.181 (−0.244, −0.118) | −0.102 (−0.175, −0.03) | **−0.147 (−0.195, −0.1)** | 0.96 |
| | TLS vs. control | 6 | −0.086 (−0.127, −0.046) | 0.014 (−0.114, 0.143) | **−0.077 (−0.116, −0.038)** | 0.98 |
| Products of mixed healthfulness | NS vs. control | 2 | 0.066 (−0.026, 0.157) | . | 0.066 (−0.026, 0.157) | 1.00 |
| | NS vs. TLS | 0 | . | 0.108 (−0.01, 0.226) | 0.108 (−0.01, 0.226) | 0 |
| | TLS vs. control | 2 | −0.043 (−0.118, 0.032) | . | −0.043 (−0.118, 0.032) | 1.00 |
| More healthful products | HW vs. control | 2 | −0.016 (−0.128, 0.097) | . | −0.016 (−0.128, 0.097) | 1.00 |
| | HW vs. NS | 0 | . | −0.137 (−0.319, 0.045) | −0.137 (−0.319, 0.045) | 0 |
| | HW vs. NW | 0 | . | 0.064 (−0.113, 0.241) | 0.064 (−0.113, 0.241) | 0 |
| | HW vs. TLS | 0 | . | −0.081 (−0.215, 0.053) | −0.081 (−0.215, 0.053) | 0 |
| | NS vs. control | 2 | 0.121 (−0.022, 0.264) | . | 0.121 (−0.022, 0.264) | 1.00 |
| | NS vs. NW | 0 | . | 0.201 (0.003, 0.399) | **0.201 (0.003, 0.399)** | 0 |
| | NS vs. TLS | 0 | . | 0.056 (−0.104, 0.216) | 0.056 (−0.104, 0.216) | 0 |
| | NW vs. control | 1 | −0.069 (−0.222, 0.084) | −0.125 (−0.437, 0.186) | −0.08 (−0.217, 0.057) | 1.00 |
| | NW vs. TLS | 1 | −0.156 (−0.309, −0.003) | −0.1 (−0.411, 0.212) | **−0.145 (−0.282, −0.008)** | 1.00 |
| | TLS vs. control | 6 | 0.066 (−0.006, 0.139) | . | 0.065 (−0.007, 0.138) | 1.00 |

RMD refers to the percentage of change comparing intervention with control group (RMD = (x2 − x1) / x1; x1: mean of continuous outcome in the intervention group or after implementation of intervention, x2: mean of continuous outcome in the control group or before implementation of intervention).

CI, confidence interval; HW, health warning; NS, Nutri-Score; NW, nutrient warning; RMD, relative mean difference; TLS, traffic light labelling system.

especially in populations with low socioeconomic status and limited knowledge of nutrition labels [24,54]. The health communication theory suggested that attention to the FOPL is a prerequisite for establishing specific perception of the label itself and an understanding of the nutrition information. Our study, together with previous evidence [23,24,30], suggested that colour-coded labels and warning labels were able to ease the difficulty in processing nutrition information and improve the objective understanding of nutrition information with the use of eye-catching design, but the perception of labels might differ, especially between the 2 colour-coded labels (TLS and NS). Although TLS and NS both used colour scales to indicate healthfulness of foods, TLS scores the level of each target nutrient, while NS summarises the overall nutritional quality taking all preferable and detrimental nutrients into consideration. Therefore, according to our study, TLS was perceived to provide sufficient information, which was also reflected in the finding that TLS was associated with a better performance in complex understanding tasks (e.g., classification of sugar and saturated fat). However, too much

**Table 5. The network effects of different label types on the perceived disease risk of consuming products.**

| Outcome | Comparison | Number of comparisons | Direct estimate (RMD and 95% CI) | Indirect estimate (RMD and 95% CI) | Network estimate (RMD and 95% CI) | Proportion of direct evidence |
|---|---|---|---|---|---|---|
| Less healthful products | HW vs. control* | 10 | **0.124 (0.053, 0.196)** | 0.6 (0.153, 1.048) | 0.136 (0.065, 0.207) | 1.00 |
| | HW vs. NW* | 2 | 0.063 (−0.09, 0.215) | −0.562 (−0.766, −0.357) | −0.16 (−0.283, −0.038) | 0.67 |
| | HW vs. NW + HW* | 1 | 0.03 (−0.181, 0.241) | −0.653 (−1.006, −0.3) | −0.15 (−0.331, 0.031) | 0.93 |
| | NW vs. control* | 3 | **0.43 (0.299, 0.561)** | −0.304 (−0.582, −0.027) | 0.296 (0.178, 0.415) | 0.66 |
| | NW vs. NW + HW* | 1 | −0.11 (−0.324, 0.104) | 0.463 (0.048, 0.878) | 0.01 (−0.18, 0.2) | 0.61 |
| | NW + HW vs. control | 1 | 0.36 (0.146, 0.574) | 0.102 (−0.236, 0.439) | **0.286 (0.105, 0.467)** | 0.23 |
| More healthful products | HW vs. control | 2 | 0.024 (0.008, 0.04) | . | **0.024 (0.008, 0.04)** | 1.00 |

RMD refers to the percentage of change comparing intervention with control group (RMD = (x2 − x1) / x1; x1: mean of continuous outcome in the intervention group or after implementation of intervention, x2: mean of continuous outcome in the control group or before implementation of intervention).

*The direct and indirect effects were observed significantly inconsistent ($p < 0.05$), and the network estimate may violate the assumption of consistency and transitivity of NMA, thus only direct evidence was used for interpretation.

CI, confidence interval; HW, health warning; NMA, network meta-analysis; NS, Nutri-Score; NW, nutrient warning; RMD, relative mean difference; TLS, traffic light labelling system.

information could also elicit confusion in consumers when reading TLS. NS, on the other hand, was perceived to be more salient and thus thought easier to understand for consumers.

According to the health communication theory and previous studies [22–26], mechanisms of motivating consumers' behavioural change vary across FOPL types [26]. NW and HW, compared with the colour-coded counterparts, are more dependent on eliciting the perception of severe risk and negative emotions, which mediate the reduced selection of unhealthful products [26]. TLS and NS, on the other hand, rely more on enhancing the perception of healthfulness for more healthful options and thereby perform better at promoting purchase of healthful foods [27,28,31], which were supported by our findings (Tables 4 and 5). Our study, together with previous evidence, explained the role of perception of healthfulness in mediating the effect of colour-coded labels and warning labels on consumers' purchasing behaviour. Warning labels are more often associated with "danger" due to the use of symbols (e.g., octagon stopping sign), colour (black and white), and cautionary texts, thus elicit negative perception and emotions towards unhealthful food products marked with warnings on the front of packages. For healthful products, warnings are not displayed, but TLS is presented with green lights on, which is indicative of the concept of "health," "nature," and "sustainability," thus associated with perception of better healthfulness for food products.

While our review found that FOPL can effectively change purchasing behaviour, mandatory FOPL policies are likely to be much more effective at changing consumption than voluntary policies. Voluntary FOPL systems have been adopted slowly in the marketplace, and consumers also perceived the products without FOPL as more healthful, albeit the nutritional quality might be worse [15]. Even though most FOPL policies are currently implemented on a voluntary basis, over the past 3 years mandatory FOPL systems have increasingly been favoured, of which warning labels and TLS were the most popular interpretative FOPLs worldwide. More countries have also proposed to make ongoing voluntary FOPL policies mandatory, which would better guide consumers' food choice and stimulate reformulation in the food industry [7,8]. To ensure the effectiveness of mandatory FOPL policies, consumer education and monitoring systems for the market should be launched, and further studies should

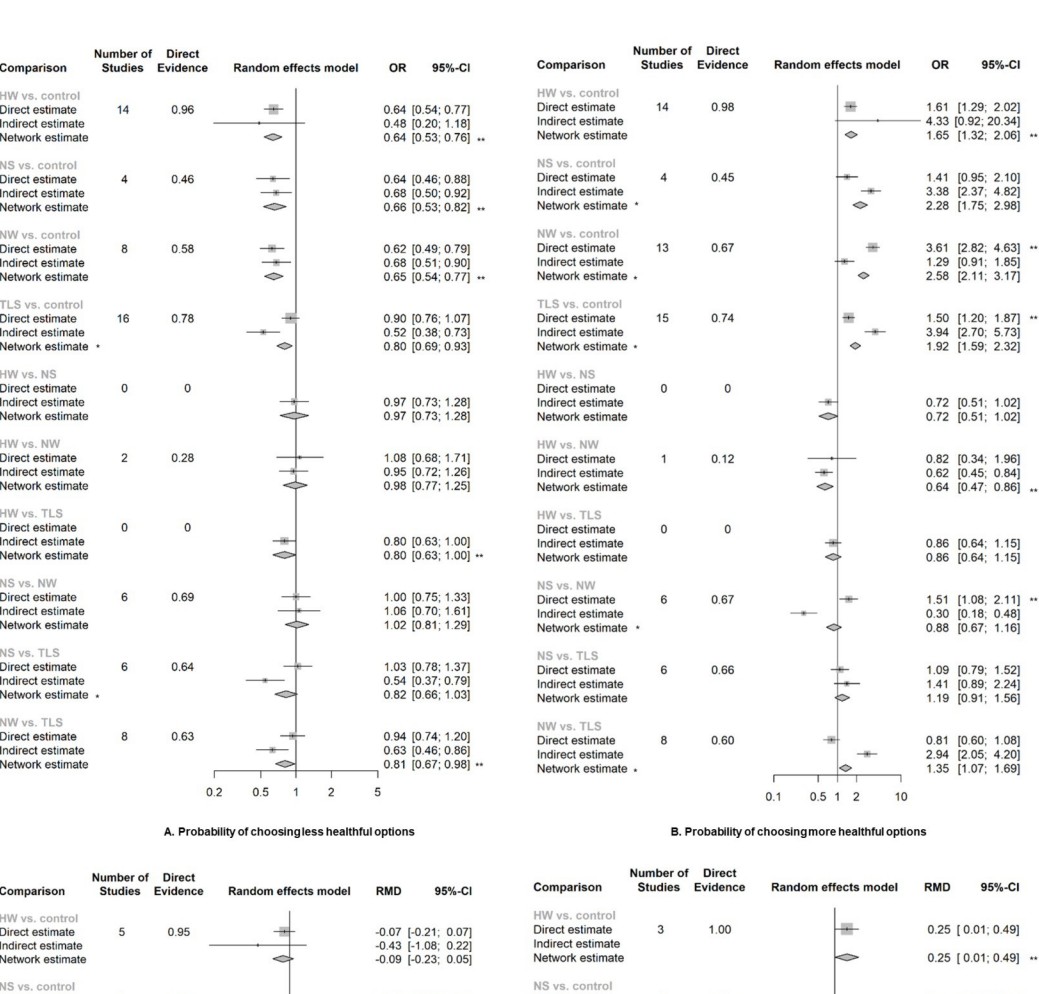

A. Probability of choosing less healthful options

B. Probability of choosing more healthful options

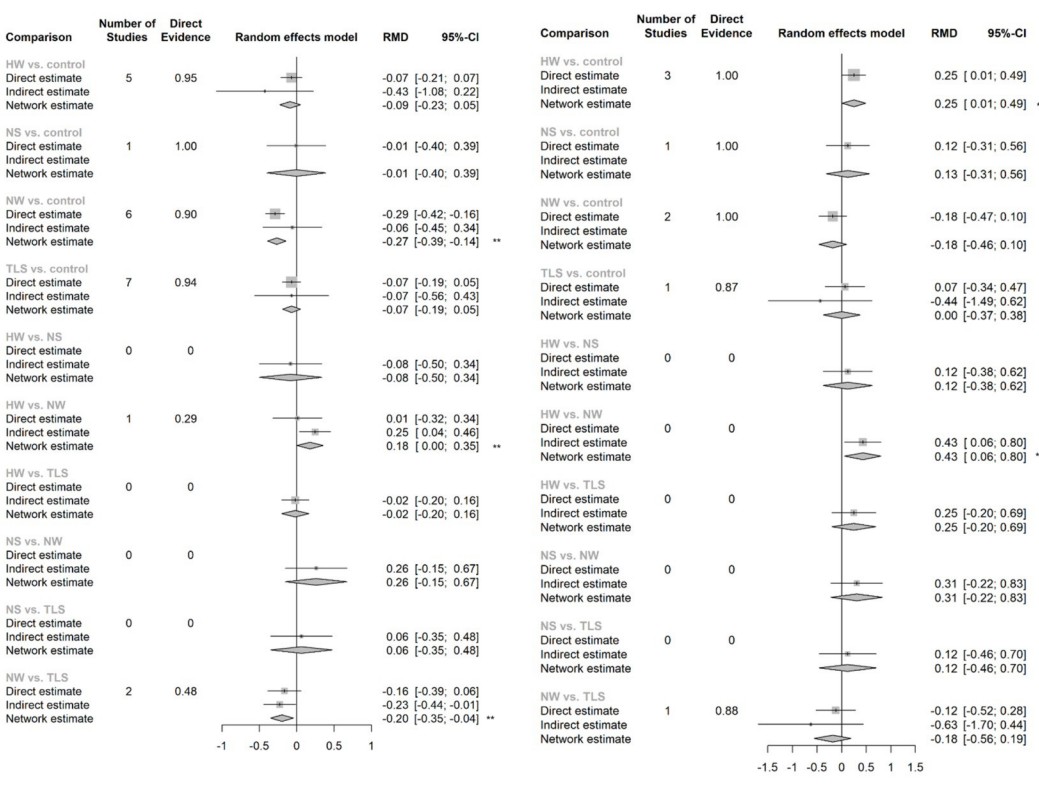

C. Purchase intention for less healthful options

D. Purchase intention for more healthful options

**Fig 4. The network effects of different label types on the purchasing behaviour or self-reported purchase intention of products.** RMD refers to the percentage of change comparing intervention with control group (RMD = (x2 − x1) / x1; x1: mean of continuous outcome in the intervention group or after implementation of intervention, x2: mean of continuous outcome in the control group or before implementation of intervention). *The direct and indirect effects were observed significantly inconsistent ($p < 0.05$), and the network estimate may violate the assumption of consistency and transitivity of NMA, thus only direct evidence was used for interpretation. **Estimates with significant effect given $\alpha = 0.05$ and $\beta = 80\%$. CI, confidence interval; HW, health warning; NMA, network meta-analysis; NS, Nutri-Score; NW, nutrient warning; RMD, relative mean difference; OR, odds ratio; TLS, traffic light labelling system.

also be carried out in real-world settings to add to the evidence of mandatory FOPL policies in different populations and regional cultures.

Compared to previous meta-analyses that assessed the effectiveness of colour-coded labels and warning labels against a control group (mostly no labels) [25,31,41,46], we utilised the

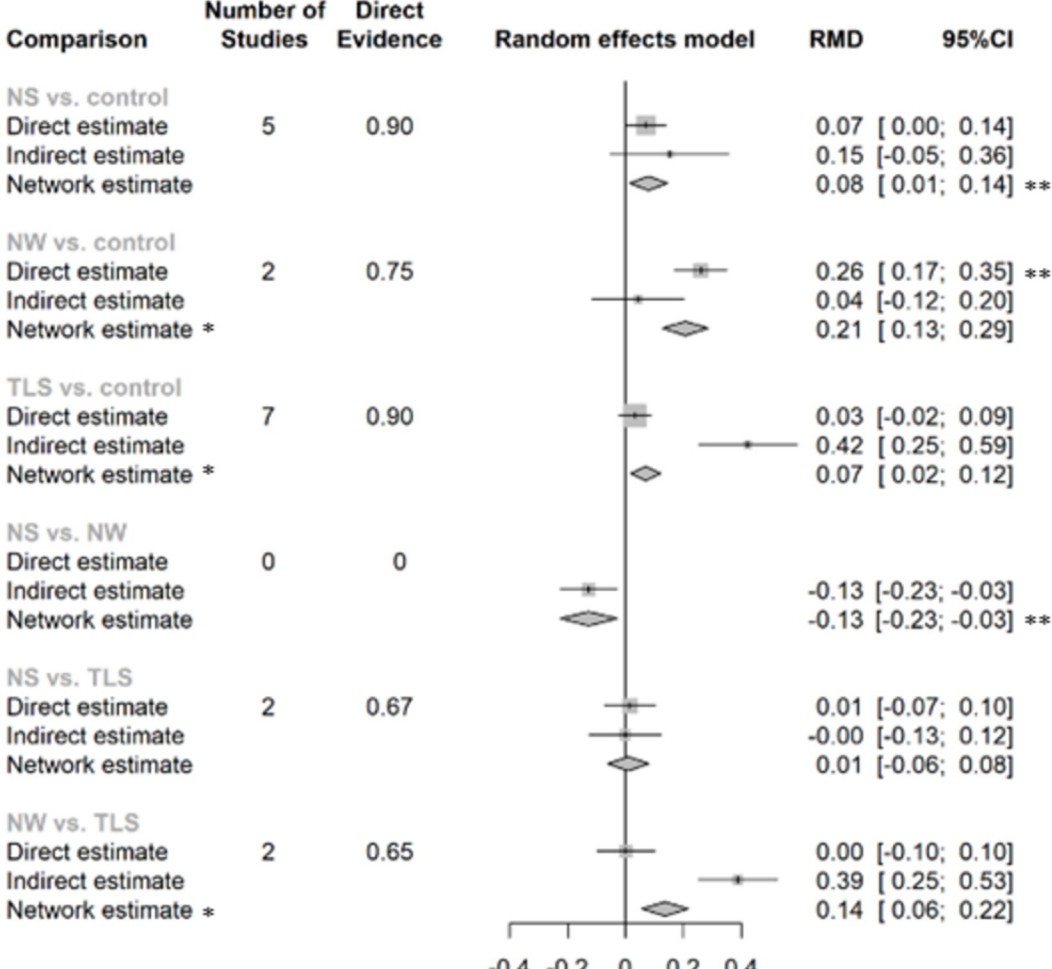

**Fig 5. The network effects of different label types on the overall healthfulness purchased/chosen (RMD and 95%CI).** RMD refers to the percentage of change comparing intervention with control group (RMD = (x2 − x1) / x1; x1: mean of continuous outcome in the intervention group or after implementation of intervention, x2: mean of continuous outcome in the control group or before implementation of intervention). *The direct and indirect effects were observed significantly inconsistent ($p < 0.05$), and the network estimate may violate the assumption of consistency and transitivity of NMA, thus only direct evidence was used for interpretation. **Estimates with significant effect given $\alpha = 0.05$ and $\beta = 80\%$. CI, confidence interval; HW, health warning; NMA, network meta-analysis; NS, Nutri-Score; NW, nutrient warning; RMD, relative mean difference; TLS, traffic light labelling system.

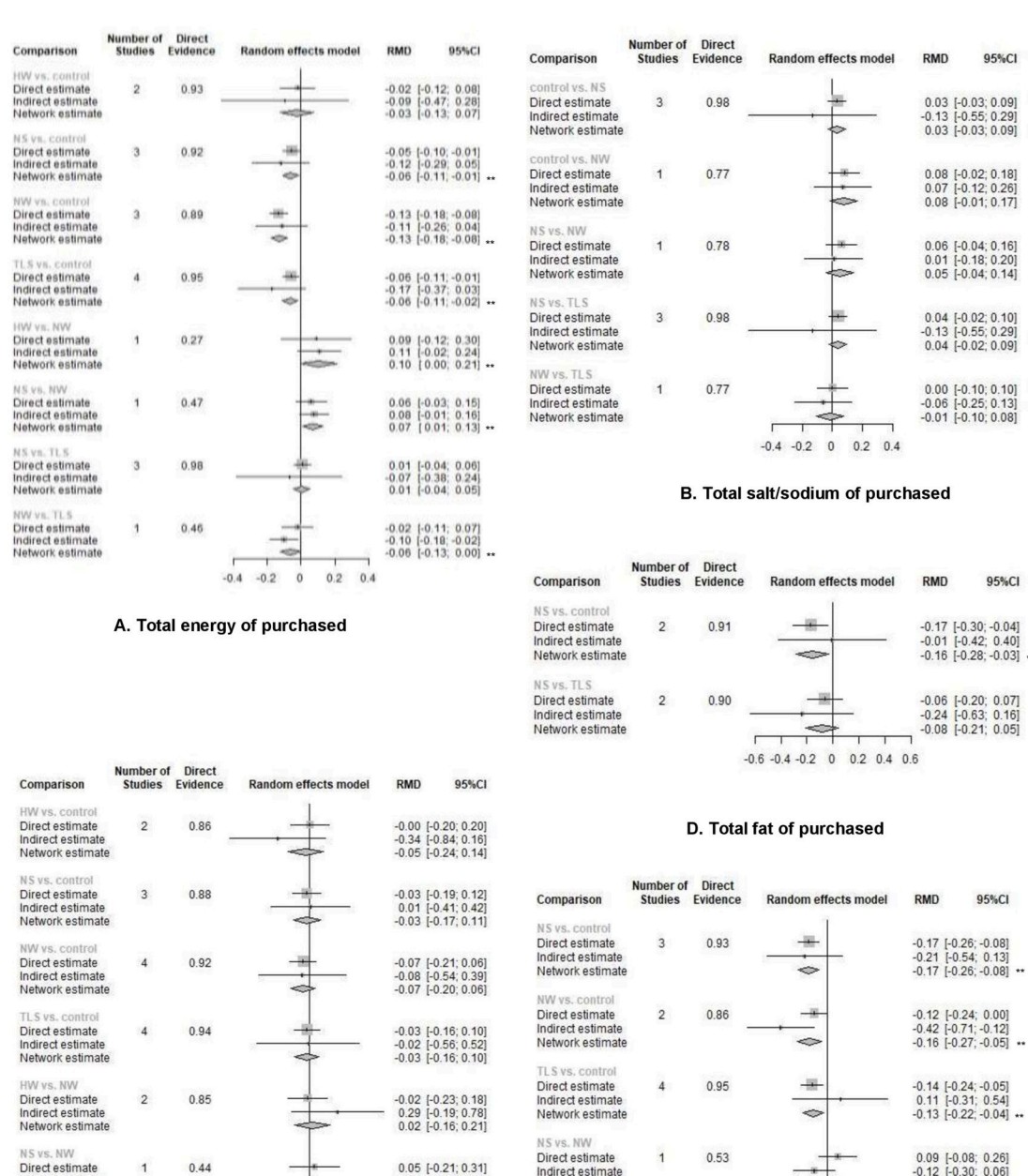

**Fig 6. The network effects of different label types on the total energy or single nutrient content purchased/chosen (RMD and 95% CI).** RMD refers to the percentage of change comparing intervention with control group (RMD = (x2 − x1) / x1; x1: mean of continuous outcome in the intervention group or after implementation of intervention, x2: mean of continuous outcome in the control group or before implementation of intervention). **Estimates with significant effect given α = 0.05 and β = 0.05. CI, confidence interval; HW, health warning; NS, Nutri-Score; NW, nutrient warning; RMD, relative mean difference; TLS, traffic light labelling system.

NMA method to explore the difference in the effectiveness and indicators of psychological mechanisms of one type of FOPL compared to another. This provided a deeper insight into the question of which is the optimal FOPL that can be applied to each country. Generally, our findings suggested that different FOPL types might dominate different outcome measures. NS performed better in nudging the purchase of more healthful products than NW, while NW had the advantage in discouraging unhealthful purchasing behaviour. The underlying psychological mechanisms also vary across labels. Warning labels reduced the perceived healthfulness of unhealthful products and remind consumers to eat less of unhealthful foods, while colour-coded labels enhanced the perception of healthfulness for more healthful products (Table 4 and S4 Table). Consumers' perception towards labels differed across types as well. NW was recognised as easier to understand and improved the classification of nutrient content (e.g., sugar and saturated fat) compared to TLS. According to our NMA, TLS was considered to provide more sufficient information than NS, but the latter was recalled more frequently (see Results in the "Salience and visual attention" section). The performance of colour-coded labels and warning labels on multiple dimensions make it necessary for policy-makers to weigh pros and cons according to local context. For example, in countries with high levels of NCDs, which place a large burden on individuals and healthcare systems, and a food system dominated by ultraprocessed food and drinks, NW or HW could help lower supply-driven consumption as they are known to be easier to interpret by consumers, who can recognise that they are applied to unhealthful products high in sugar, salt, and saturated fat [55,56].

Our review included empirical evidence from multiple databases, providing the latest and most comprehensive evidence for different aspects of colour-coded and warning labels, including different subtypes: summary and nutrient-specific type, nontextual and textual types, and positive- and negative-framed types. We applied the NMA method to make full use of both direct and indirect evidence of the comparisons between intervention labels and the control condition. These results can support policy-makers to make decisions based on the performance of labelling in different dimensions. In addition to the effect on purchasing and consuming behaviour, we also investigated the underlying psychological outcomes quantitatively based on the health communication framework, which systematically suggested the mechanism underpinning the effect of consumers' behaviour.

Our study had some limitations. First, compared to the relatively large amount of evidence on the purchasing behaviour elicited by FOPLs, the data on food consumption were quite limited. Our findings suggested that both warning labels and colour-coded labels would reduce the perceived recommended consumption amount or frequency of unhealthful products, and warning labels might outperform colour-coded, which could be seen as an indirect evidence that FOPLs are able to change dietary consumption. However, the research gap between purchase and actual intake of different nutrients remains to be validated by future studies, which is crucial to inform decision-making on labelling policies. Second, most of the studies were laboratory experiments, and, thus, findings mainly indicate the immediate or short-term effect of colour-coded and warning label interventions. There were very few real-world studies assessing the effect of mandatory labelling policies on genuine "purchase." However, considering the highly controlled condition in RCTs and quasi-experimental studies, our findings will need further demonstration by real-world evidence to build the evidence for the generalisation of labelling to other parts of the world. In our meta-analysis, we found that most of the existing real-world studies evaluated TLS and found it effectively increased the purchase/selection of more healthful products. For NS, NW, and other HW types, more research in real-world settings is needed to confirm their effectiveness. Third, more than half of the studies included had a high risk of bias. Considering the nature of nutrition labelling intervention studies, it was inevitable that participants would be aware of their assigned interventions, and, in turn, such awareness

could influence the assessment of outcomes. Therefore, in the sensitivity analyses, we included only RCTs in the analyses of consumers' behaviour, and the results were consistent with those of the primary analyses. Fourth, we only searched for peer-reviewed articles in 4 of the most commonly used databases and the bibliographic references of eligible articles to ensure that our search strategy could be easily replicated and repeated for the update. We believed that most relevant studies should have been covered in this way, though some relevant studies reported in other databases or in the grey literature may have been missed. Fifth, we did not conduct meta-regression to explore the heterogeneity and inconsistency between direct and indirect comparisons, due to the limited number of studies for most outcomes. Instead, we used a random-effect inconsistency model to accommodate the inconsistency and heterogeneity within and across comparisons. We also carried out a series of subgroup analyses by a range of effect modifiers based on previous studies, including age, sex, study setting, accessibility of NFt, and product types. Another sensitivity analysis was also performed excluding studies using NFt as control setting. The results of the sensitivity analyses did not differ much from our findings in the primary analysis, which suggested the biases generated from combining 2 control settings (NFt control and no-label control) might be relatively small. Sixth, we only considered energy and unfavourable nutrients (e.g., sugar, salt, fat, and saturated fat) that are common in various FOPLs and are considered the major risk factors of NCDs burden [55], but favourable nutrients (e.g., fibre, protein) are also components of interest in many FOPLs (e.g., NS and Health Star Rating) for their beneficial health effect [57,58]. So far, few studies have evaluated the effect of FOPLs on favourable nutrients [59], and there has been a disagreement in the inclusion of favourable nutrients in FOPLs as they might have a health halo effect to products that are high in salt, sugar, or fat [60,61]. For these reasons, we did not summarise the results on unfavourable nutrients in this systematic review, and further studies are needed to clarify the pros and cons of favourable nutrients on colour-coded and warning labels. Finally, the numbers of studies were limited for some comparisons (e.g., only one study provided direct evidence on the comparison between NS and HW for the probability of purchasing more healthful products), especially in the analysis of outcomes concerning the perception and understanding of FOPLs, and perception and attitudes towards food options. Our findings for these secondary outcomes need to be validated by more studies in the future.

In summary, our findings suggest that both colour-coded labels and warnings appeared effective in nudging consumers' behaviour towards more healthful products by changing the healthfulness perception and eliciting negative emotions. Each type of label may have some different attributes, but the difference between different forms of labels remains to be demonstrated by further studies. Our study can support policy-makers to push forward mandatory FOPL policies to make use of the full potential of FOPL in directing consumers' food choice and encouraging reformulation in the food industry.

## Supporting information

**S1 Text. Supplementary methods.**
(DOCX)

**S1 PRISMA Checklist. NMA checklist of items to include when reporting a systematic review involving a network meta-analysis.**
(DOCX)

**S1 Fig. Forest plots of network estimates combining direct and indirect effects for consumers' perception consumers' perception and attitudes for foods and drinks (secondary outcomes).** CI, confidence interval; HW, health warning; MD, mean difference; NS, Nutri-Score;

NW, nutrient warning; OR, odds ratio; TLS, traffic light labelling system.
(TIFF)

**S2 Fig. Forest plots of network estimates for consumers' objective understanding of colour-coded and warning labels (secondary outcomes).** CI, confidence interval; NS, Nutri-Score; NW, nutrient warning; OR, odds ratio; TLS, traffic light labelling system.
(TIFF)

**S3 Fig. Forest plots of network estimates for consumers' subjective understanding, attention, and perception of colour-coded and warning labels (secondary outcomes).** CI, confidence interval; HW, health warning; MD, mean difference; NFt, Nutrition Facts table; NS, Nutri-Score; NW, nutrient warning; OR, odds ratio; TLS, traffic light labelling system.
(TIFF)

**S4 Fig. Comparison-adjusted funnel plots for comparison of interventions with control condition for changes in consumers' purchasing and consuming behaviour (primary outcomes).** HW, health warning; MD, mean difference; NS, Nutri-Score; NW, nutrient warning; TLS, traffic light labelling system.
(TIFF)

**S5 Fig. Comparison-adjusted funnel plots for changes in consumers' perception and attitudes for foods and drinks (secondary outcomes).** HW, health warning; MD, mean difference; NS, Nutri-Score; NW, nutrient warning; OR, odds ratio; TLS, traffic light labelling system.
(TIFF)

**S6 Fig. Comparison-adjusted funnel plots for consumers' attention, perception, and understanding of colour-coded and warning labels (secondary outcomes).** HW, health warning; NS, Nutri-Score; NW, nutrient warning; OR, odds ratio; TLS, traffic light labelling system.
(TIFF)

**S1 Table. Measurement and definition of outcomes in the systematic review of colour-coded and warning labels.**
(XLSX)

**S2 Table. Characteristics of studies included in the systematic review.**
(PDF)

**S3 Table. The netsplitting of network meta-analysis estimates for changes in consumers dietary behaviour into the contribution of direct and indirect evidence and test for local inconsistency.**
(XLSX)

**S4 Table. The netsplitting of network meta-analysis estimates for changes in consumers' perception and attitudes for products into the contribution of direct and indirect evidence and test for local inconsistency.**
(XLSX)

**S5 Table. The netsplitting of network meta-analysis estimates for consumers' attention, perception, and understanding towards labels into the contribution of direct and indirect evidence and test for local inconsistency.**
(XLSX)

**S6 Table. The netsplitting of network meta-analysis estimates into the contribution of direct and indirect evidence and test for local inconsistency in the analysis of label subtypes.** (XLSX)

**S7 Table. Netsplitting of network meta-analysis estimates for the primary outcomes into the contribution of direct and indirect evidence and test for local inconsistency grouped by age, sex, SES, study setting, type of products, and NFt display.** (XLSX)

**S8 Table. The netsplitting of sensitivity network meta-analysis in studies estimates (including only randomised controlled trials) for changes in consumers dietary behaviour into the contribution of direct and indirect evidence and test for local inconsistency.** (XLSX)

**S9 Table. The netsplitting of sensitivity network meta-analysis estimates (removing studies using NFt control) for changes in consumers dietary behaviour into the contribution of direct and indirect evidence and test for local inconsistency.** (XLSX)

**S10 Table. Risk of bias of randomised controlled trials included in the systematic review using the revised Cochrane risk-of-bias tool for randomised trials (RoB 2).** (XLSX)

**S11 Table. Risk of bias of quasi-experiments included in the systematic review using Risk Of Bias In Non-randomised Studies of Interventions (ROBINS-I).** (XLSX)

**S12 Table. Risk of bias of cross-sectional studies included in the systematic review using NHLBI study quality assessment tools.** (XLSX)

**S1 Data. Details of included studies.** (XLSX)

## Author Contributions

**Conceptualization:** Jing Song, Mhairi K. Brown, Monique Tan, Jacqui Webster, Norm R. C. Campbell, Kathy Trieu, Cliona Ni Mhurchu, Laura K. Cobb, Feng J. He.

**Data curation:** Jing Song, Mhairi K. Brown, Feng J. He.

**Formal analysis:** Jing Song, Mhairi K. Brown.

**Methodology:** Jing Song.

**Visualization:** Jing Song.

**Writing – original draft:** Jing Song, Mhairi K. Brown.

**Writing – review & editing:** Jing Song, Mhairi K. Brown, Monique Tan, Graham A. MacGregor, Jacqui Webster, Norm R. C. Campbell, Kathy Trieu, Cliona Ni Mhurchu, Laura K. Cobb, Feng J. He.

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
