## [Editor Report · Decision Letter 0]

11 Dec 2020

Dear Dr He, 

Thank you for submitting your manuscript entitled "Impact of color-coded and warning nutrition labelling schemes: a systematic review and network meta-analysis" for consideration by PLOS Medicine.

Your manuscript has now been evaluated by the PLOS Medicine editorial staff and I am writing to let you know that we would like to send your submission out for external peer review.

Sincerely,

Richard Turner, PhD,

Senior editor, PLOS Medicine

rturner@plos.org

---

## [Decision Letter · Decision Letter 1]

11 Jan 2021

Dear Dr. He,

Thank you very much for submitting your manuscript "Impact of color-coded and warning nutrition labelling schemes: a systematic review and network meta-analysis" (PMEDICINE-D-20-05962R1) for consideration at PLOS Medicine. 

[LINK]

In light of these reviews, I am afraid that we will not be able to accept the manuscript for publication in the journal in its current form, but we would like to consider a revised version that addresses the reviewers' and editors' comments. Obviously we cannot make any decision about publication until we have seen the revised manuscript and your response, and we plan to seek re-review by one or more of the reviewers. 

We expect to receive your revised manuscript by Feb 01 2021 11:59PM. Please email us (plosmedicine@plos.org) if you have any questions or concerns.

We look forward to receiving your revised manuscript. 

Sincerely,

Emma Veitch, PhD

PLOS Medicine

On behalf of Richard Turner, PhD, Senior Editor, 

PLOS Medicine

plosmedicine.org

*Please restructure the abstract using the PLOS Medicine headings (Background, Methods and Findings, Conclusions - "Methods and Findings" is a single subsection).

*In the Abstract, following text, assume that the OR's given in the brackets reflects the range of OR's for the point estimates for those specific four forms of displaying nutrition information? This is a somewhat unusual way of representing statistics for this type of study, where generally you'd expect an OR for the point estimate followed by 95% CIs pertaining to that point estimate. Some readers might find this presentation a bit confusing, so if the authors can use a more conventional presentation that might be helpful. 

"We found that the Traffic-light labelling system (TLS), Nutri-score (NS), Nutrient Warning (NW) and Health

Warning (HW) all increased the probability of selecting healthier products (OR: 1.71-2.83), and warning labels were particularly effective in reducing consumers’ probability of selecting less healthy products (odds ratio [OR]: 0.61-0.62)". 

*In the abstract, for the following statement, is it possible to add information on what the outcome measure is for "healthfulness"? Also note that "healthy/unhealthy" is often seen as British English but "healthful" is more US English, would suggest the authors go for one or the other rather than a mix. 

"TLS, NS and NW increased the overall healthfulness (TLS: +3%; NS: +5.4%; NW: +3.1%)"

*In the last sentence of the Abstract Methods and Findings section, please add a note about any key limitation(s) of the study's methodology.

*In the author summary, some of the bullets could be rephrased a bit to flow better:

- Evidence on the impact of each type of color-coded labels and warning labels on modifying consumers purchasing behaviours were mixed. - perhaps "Prior to this study, evidence on the impact of...... was mixed"

- The feasibility and effectiveness of each label type applied in different contexts were

unclear. - perhaps "Previously, the feasibility and likely effectiveness of using different label types in different contexts was unclear".

*Suggest also rephrase the following:

- We provided the most comprehensive evidence to guide policy-makers in choosing the optimal front-of-package labelling policies, which will also inspire the generalization of mandatory front-of-package labelling schemes and largely mitigate the noncommunicable chronic disease burden. - perhaps tone this down a bit - "We provide more comprehensive evidence to guide..... This evidence synthesis may inform further generalization of mandatory front-of-package labelling schemes and help to mitigate the burden of noncommunicable chronic disease"

*Suggest perhaps the following bullet could instead be replaced by something noting any limitations of the study which might affect interpretation/generalisation? 

- In the future, we will expand our network meta-analysis work by summarizing on the impact of interpretative front-of-package labelling schemes on industrial reformulation and consumers’ actual consumption behaviours.

*Please reformat the citation style into PLOS Medicine's format (should be straight forward if using referencing software) - this should use callouts formatted as sequential numerals in square brackets (not superscript).

Comments from the reviewers:

Reviewer #1: The manuscript deals with an actual and highly relevant topic for policy making worldwide. The authors have done a tremendous amount of work summarizing and analyzing all the available information on the topic. However, some of the analyses are not clearly explained or presented, which makes it difficult to judge the conclusions reached by the authors. I recommend a major revision after major methodological issues are addressed. Based on the data report on the paper I cannot say whether results are valid or not. Please find below some detailed comments. When submitting a revised version of the manuscript, please include line number to facilitate the review.

Introduction

- The manuscript should include a logic model explaining how front-of-package nutrition labelling (FOPNL) is expected to influence health outcomes. References to the health communication theory should be provided on the Introduction.

- The authors state that "Observational studies found that the mandatory implementation of warning labels in Chile resulted in lower sales of beverages high in sugars, salt, saturated fat or energy". However, the Chilean law does not only include FOPNL, which makes impossible to disentangle the effects of the different components (FOPNL, marketing regulations and prohibition of products with warnings at schools). In addition, Study 21 only reports qualitative data from focus groups with a specific group of participants, so the authors should tune down their statements related to the mechanisms underlying the efficacy of the warnings.

Methods

- The authors should include detailed information on the criteria used for assessing risk of bias. This is particularly relevant for risk of bias for measurement of the outcome and confounding, given that most studies were categorized as high risk of bias. The authors refer to Table S22 but do not provide any specific explanation of how it was assessed or the reasons underlying the classification.

- Studies on FOPNL have been conducted using very different methodologies and study designs. Therefore, it is not clear how the authors calculated and analyzed the data to report ORs for all the variables. This is a major concern given that conclusions are reached based on these estimates. A detailed description on how these ORs were calculated should be provided in the manuscript. 

- A similar comment applies to the network analysis, more detailed information should be provided to enable reviewers to actually judge the validity of the conclusions. 

Results

- The authors have included a huge number of Tables and Figures, which makes it difficult to follow the manuscript. I recommend the authors to reduce the number of Tables and Figures to facilitate understanding.

- In Table 1, I recommend providing a more detailed split of the publication year of the articles. The authors grouped 2018-2020 and 1998-2018 without a clear rationale. Regarding countries, I think that summary information by continent or region of the world will be more informative than country income. For example, Chile and Uruguay are high income countries but are very different from European countries or the USA. Regarding SES, I do not agree with how the authors grouped studies as they may be using different criteria than the ones reported in the articles. They should refrain from re-categorizing variables reported in the studies. Finally, considering that most articles do not report race, I would remove this criteria.

- Results for the different outcomes should be presented following the order of a logic model, e.g. consumer perceptions and attitudes come before purchase intention.

- The authors should clarify how "less healthy" and "healthier" were defined. This is particularly relevant given that studies tend to use different criteria. 

- Results from network analysis are not clearly presented.

- The sections consumer's perception and attitudes and consumers' understanding are actually very important but results are not presented in detail. The authors should choose between excluding these sections or providing additional information. This is particularly relevant given that these outcomes could define decisions to implement different FOPNL schemes. In their discussion section they say that the review evaluated the impact of color-coded and warning labels on "the psychological mechanism underpinning purchasing modification, including consumers' perceptions and liking for foods and drinks, as well as the understanding and evaluation of label attributes". However, detailed information is not provided. 

Discussion

- The discussion section should be rewritten and improved to provide based on an in-depth discussion of the results reported in the manuscript. In the current version, the authors discuss results that are not presented in the manuscript, e.g. "Nutrition labels with numerical information and technical terms ... added to the difficulty in understanding nutrition information". In addition, it is not clear how the authors draw some of their conclusions based on their results, e.g. "TLS is considered to provide more sufficient information than NS, but the latter is recalled more frequently"

- For the interpretation of results, the psychological mechanisms underlying the different schemes have not been adequately discussed. In this sense, recent studies showing how framing effects influence the effect of FOPNL on consumers' choice or the potential role of heurisitics on consumers' perception of products with warnings are not included in the review nor discussed. 

- Some of the comments are not entirely accurate, particularly for a review on FOPNL. For example, the authors state "In addition, ultraprocessed foods and drinks, which are usually high in free sugar, total fat, saturated fat and sodium, are generally labeled with a NW for at least one nutrient, but some of which may be labeled as healthy in TLS or NS (e.g., sugar sweetened beverages may have green lights on sodium and saturated fat contents when labeled with TLS)". I don't think it is accurate to say that the TLS would label sugar sweetened beverages as healthy, as it is not the purpose of the TLS. The problem is not the label per se, but how consumers may understand the information provided by the TLS. In this sense, the authors report one study showing this effect (Reference 61).

- The discussion of how TLS or NW may be adequate for countries with different socioeconomic status is based on experimental data. The authors state that warnings "may be more suitable for countries with high levels of citizens with low socioeconomic status, who may indicate a lack of nutrition knowledge and/or health consciousness, and who may be unable to pay for expensive healthier foods". This statement does not take into account the effect of the food environment on consumers' eating patterns. For example, the share of ultra-processed products consumed is higher in the USA or UK compared to Latin American countries. Therefore, I do not think that NW would not be effective in the USA or the UK, even if consumers may have "higher nutrition knowledge or ability to purchase healthier alternatives". The authors should revised their statements based on a food systems' approach. I recommend the review by Swinburn et al. published last year on the Lancet discussing the syndemic of obesity, undernutrition and climate change.

Reviewer #2: Feng He and colleagues presented here a systematic review and network meta-analysis about the impact of color-coded and warning nutrition labelling schemes. The topic is particularly relevant as it is a timely public health issue with high stakes in the current international debates on food nutrition labels.

The study aims to assess the impact of the color-coded and warning labels, the most studied and promising labels, on changes in both intended and actual purchasing and consumption behaviours. The study aims too to gain insight into the underlying psychological mechanism based on health communication theory to explore the heterogeneity across label types

It is an excellent and very useful paper. The manuscript is clear and well-written. The description of the methods and resultsare all good. The literature review of peer-reviewed articles on the topic (118 articles) seems to me really exhaustive. The statistical methods used are appropriate.

The study addressed interesting and very important research questions. This is a fascinating piece of work using a strong methodology. 

I have few comments. 

1. Authors claimed they followed the flow chart of the PRISMA recommendations . However it is not identical to the PRISMA flow diagram described on the website http://prisma-statement.org/. Some information are missing : how many unique articled authord have found (excluding duplicates), or how many articles were obtained from other sources (for example, by reviewing the bibliographic references of the identified articles). 

2. In their flow chart, authors indicate that 14,785 items are excluded without giving the reason: duplicates? Do not meet the inclusion criteria? If so, what criteria?

3. Two independent reviewers reviewed the articles. What was the percentage of discrepancy? Please precise.

4. The conclusions on the objective understanding of FOP Nutrition label are questionnable . Authors conclude on the effect of the TLS but not so much NS whereas many studies performed in France, Spain, Germany, UK, the Netherlands, Bulgaria, Switzeland, Italy, Singapore, USA, Canada,… have focused on this point and show well the superiority of the NS...it is visible in the additional figure S4 and the difference in outcome between Figure A and C is not understandable...

5. Sometimes discussion is not consistent with results. In some places, authors highlight the performances of TLS in the discussion while this is true also on the NS (and it is said in the results).

6. The proposed outcomes in term of nutrients include only unfavourable nutrients (I agree this is a major point) but they do not include favourable nutrients, while it is a major point of interest for NS (and other FOP nutrition labels in the world such as australian HSR) to have effects on positive components, especially +++ fiber. It would be worthwhile to point out that issue, since the effects of FOP nutrition labels must be able to take into account all nutrients which could have an effect on health.

7. Authors should include in their discussion the paper on serving sizes from Egnell et al (Egnell M, Kesse-Guyot E, Galan P, Touvier M, Rayner M, Jewell J, Breda J, Hercberg S, et Julia C. Impact of Front-of-Pack Nutrition Labels on Portion Size Selection : An Experimental Study in a French Cohort. Nutrients 10, no 9 (2018) : 1268) , especially when they talk about the 'gap in knowledge' between buying and consumption.

8. The study of Acton et al. on the NS should not be taken into considération for analysis of Nutri-Score since, as Acton et all indicated in their methodology, they used the graphic format of NS in their study, they did not used the correct algorithm for the NS (this is responsible of important modification for classification fo foods)

9. In tables 2 and 3, some limits of the interval are not accurate (either they are significant and should be bold, or they are not and should not be ( I think it is a rounding problem). For ex : NW vs TLS (-0.051 (-0.103,0) ; TLS vs control (-0.026 (-0.051,0))

10. Under Tables 2 to 4, the word intervention has been replaced by the word interaction in the footnote

11. I did not understand what the N column correspond to in tables 2 to 4. I thought it was the number of studies included in the aggregation of the statistical parameter, but if it was the case, there shouldn't be zeros…. Please clarify.

Reviewer #3: 

Impact of color-coded and warning nutrition labelling schemes: a systematic review and network meta-analysis. 

Song et al

This well conducted systematic review and meta-analysis aimed to analyse the impact of color-coded interpretive labels and warning labels on consumer purchasing behaviour. 

The results are potentially important for policy makers and the wider public health community, 

as are the potential insights into the underlying psychological mechanisms of behavioral change.

ABSTRACT

Generally good.

Results, line 16

Here, and at various points beyond, please replace "showed" by "suggested". 

(Best to be a little modest and circumspect, given the limitations of the data, and analyses).

INTRODUCTION

Line 6 "To mitigate the healthcare burden resulting from NCDs, providing clear information about the nutritional profile of products is a recognised method to nudge consumers to healthier food

and drink options 8." 

Please also mention that nutrition labelling also puts pressure on manufacturers to reformulate. Currently that important concept is not mentioned until late in the manuscript (Page 19). "..stimulate reformulation in the food industry53,54."

Para 1, penultimate line "systems, such as the Guideline Daily Amount, convey nutritional content, allowing consumers …", "

Please say "systems, such as the Guideline Daily Amount, convey nutritional content as numbers rather than graphics, allowing consumers …",

METHODS

The protocol of this systematic review was registered on PROSPERO (CRD42020161877). - Good

Search Strategy. They searched four databases. 

Did they do any hand-searching, or consultation of topic-experts?? If not, they likely missed some studies.

They sensibly identified one Primary Outcome: 

changes in consumers' purchasing and consumption behaviours, 

and relegated two other outcomes to Secondary status: a) consumer's perceptions and attitudes towards products, and b) consumers' understanding and perceptions of color-coded and warning labels.

P9. "Inclusion and exclusion, data extraction and risk of bias were first assessed by one reviewer

(J.S), independently reviewed by a second reviewer (M.B)". Good.

RESULTS

Generally good.

Most studies were carried out in a laboratory setting (94%). Obvious issues of generalisability to real world settings. This is picked up in the Discussion.

74% had a high risk of bias.

What % were industry funded?

Tables 2 & 3 are particularly informative, but complicated.

Might it be possible to present the key findings as a graphic, perhaps a Forrest Plot?

The language throughout the manuscript needs to be toned down, and made a bit less categorical

Eg Abstract 

"Results. A total of 135 studies nested in 120 articles were incorporated into the systematic review, of which 118 studies in 105 articles were eligible for meta-analysis.

We found that the Traffic-light labelling system (TLS), Nutri-score (NS), Nutrient Warning (NW) and Health Warning (HW) all INCREASED the probability of selecting healthier products (OR: 1.71-2.83),…"

Given the limitations of the data, and analyses, it would be wiser to say:

"We found that the Traffic-light labelling system (TLS), Nutri-score (NS), Nutrient Warning (NW) and Health Warning (HW) APPARENTLY all increased the probability of selecting healthier products (OR: 1.71-2.83),…"

Likewise, for intance, in the main Results, Page 16, line 2:

"When color-coded labels and warning labels were compared against each other, we found that NS WAS more effective than NW in promoting the probability of purchasing healthier

products (OR and 95%CI: 1.64 [1.09, 2.48]),…"

Better to say:

"When color-coded labels and warning labels were compared against each other, we found that NS APPEARED more effective than NW in promoting the probability of purchasing healthier products (OR and 95%CI: 1.64 [1.09, 2.48]),…"

In general, almost every use of the term "showing" would be more honestly stated by using the term "suggesting". 

That would not detract at all from the authority or main messages of this comprehensive analysis.

The results sections also needs at least one sentence summarising the results of the Funnel Plot analyses.

DISCUSSION

Generally good.

Page 18 states " our study found that FOPLs had a positive effect on guiding consumers in making healthier food choices, 

especially in populations with low socio-economic status and limited knowledge of nutrition labels 8,52."

However, I cannot see any results text to support the statement about "especially in low SE status" ??

Limitations para.

Generally good. 

But it would be nice to see an acknowledgement that some some studies were probably missed, however, that would be unlikely to alter the main messages. 

Acknowledgements

It might be sensible to state how the various authors were funded (mainly to pre-empt subsequent unwarranted accusations of conflicts of interest).

Supplement

generally informative.

I would like to see the very useful Figure S5 promoted into the main manuscript;

Perhaps split into two figures, one covering the more "objective" measures of purchasing/consumption (calories, salt, fat etc), 

the other the various psychological/behavioural intentions.

The PICOS Table (Table S1 PICOS criteria for inclusion and exclusion of studies) might also be usefully promoted into the main manuscript.

Nil else.

Reviewer #4: See attachment

Michael Dewey

[LINK]

---

## [Decision Letter · Decision Letter 2]

23 May 2021

Dear Dr. He,

Thank you very much for submitting your manuscript "Impact of color-coded and warning nutrition labelling schemes: a systematic review and network meta-analysis" (PMEDICINE-D-20-05962R2) for consideration at PLOS Medicine. We do apologize the delay in sending you a response. 

Your paper was re-evaluated by our independent reviewers, including a statistical reviewer. The reviews are appended at the bottom of this email and any accompanying reviewer attachments can be seen via the link below:

[LINK]

In light of these reviews, we will again be unable to accept the manuscript for publication in the journal in its current form, but we would like to invite you to submit a further revised version that addresses the reviewers' and editors' comments fully. You will appreciate that we cannot make a decision about publication until we have seen the revised manuscript and your response, and we expect to seek re-review by one or more reviewers. 

We hope to receive your revised manuscript by Jun 11 2021 11:59PM. Please email us (plosmedicine@plos.org) if you have any questions or concerns.

Please let me know if you have any questions, and we look forward to receiving your revised manuscript. 

Sincerely,

Richard Turner, PhD

rturner@plos.org

Please update the search to the end of March, 2021, say.

Please make that "quasi-experimental studies" in the abstract and elsewhere.

Please quote the numbers of randomized and non-randomized studies included, around line 11 of the abstract.

In the abstract and elsewhere in the paper, please indicate where you are quoting findings from randomized and non-randomized studies. Where data from the latter study designs are included, please adapt the language used so as not to imply causality, e.g., "... were associated with reduced probability of consumers purchasing ...".

Please adapt the final sentence of the "Methods and findings" subsection of your abstract so that it begins "Study limitations include ..." or similar, and quotes 2-3 of the study's main limitations. 

Throughout the text, please adapt reference call-outs so that they are preceded by a space, and contain no spaces within the square brackets, e.g., "... nutrition labels [24,54].".

In the reference list, please use the journal name abbreviation "PLoS Med.".

Please break the PRISMA checklist out into a separate attached file, labelled "S1_PRISMA_Checklist" or similar and referred to as such in the text. 

In the checklist, please refer to individual items by section (e.g., "Methods") and paragraph number, not by line or page numbers, as these generally change in the event of publication. 

Comments from the reviewers:

*** Reviewer #1: 

The authors have largely improved their manuscript according to the reviewers' suggestions. However, there are several major issues that have not been adequately addressed and deserve further consideration.

- As suggested, the authors have included a logic model to explain the effect of FOPL on consumer behavior. However, the model deserves revision. Attention is a pre-requisite for perception and understanding. Intention to purchase or consume is not the same as purchase or consumption. This should be clarified in the model, even if the authors grouped the two outcomes for analysis. Additional explanations should be included in the text. The authors refer to "negative perception of food products" as a requirement to modify food choice. This is not necessarily the case. FOPL can potentially modify choice by increasing healthiness perception (e.g. health logos).Something similar occurs with references to negative emotions (Line 126). I recommend the authors to more carefully examine the literature on the effects of FOPL on consumer perception and behavior. In Line 114, the authors refer to "confounding", which is not accurate. The characteristics of the schemes, product category and personal characteristics moderate the influence of FOPL on consumer behavior. This should be clearly explained in the manuscript.

- Throughout the manuscript, the authors refer to "colour coded" FOPL and merge traffic-light with Nutriscore. However, these two schemes are conceptually different. What about the differences between the two schemes? The rationale for grouping these two schemes should be better explained and their differences further discussed.

- Grouping of the outcomes should respond to the logic model. As I previously commented, salience and attention should not be grouped with understanding and perception. The term "recall" is used for expressing different outcomes. However, this is not clear in the manuscript. For example, Table 3 refers to recall of overall healthfulness (Supplementary Table 1) but the authors only refer to "recall"

- Some of the effects reported in the Tables were calculated based on 1 or 2 comparison. This should be acknowledged as a limitation of the analysis.

- The discussion section lacks accuracy in many sections. The authors should be more careful in their description of the findings. I include below a couple of examples

Line 504: "perception of severe risk" there is no evidence to state that the labels raised associations with "perceived risk"

Line 534: "remind consumers to eat less of unhealthful foods" what is the evidence for this?

- Lines 507-511: Evidence has shown that the traffic light system does not decrease healthfulness perception as much as warnings for products with low content of some nutrients. This should be further discussed.

- Differences between TFL and Nutriscore should be discussed in depth in the discussion, as they provide different types of information.

*** Reviewer #2: 

Authors have satisfactorily taken into consideration my different comments

I think this paper is relevant and particularly important and deserves to be published. 

*** Reviewer #4: 

I still feel that combining no label and NFt is a mistake. The authors' rebuttal has not convinced me here. Their point 1 is that other people do it. That does not make it right. Their point 2 is about precision but I am concerned about bias. Their point 3 seems to suggest that they have done the wrong study. If the interest is in whether FOPLs increase the effectiveness of NFt then only those studies are relevant not the ones comparing FOPL with no label.

I am not sure where the authors got the idea that I advocate removing material from the supplement or elsewhere. My view is that most studies are under-reported and I would usually be suggesting adding more not taking away. But that is a minor point.

Michael Dewey

***

[LINK]

---

## [Decision Letter · Decision Letter 3]

5 Aug 2021

Dear Dr. He,

Thank you very much for re-submitting your manuscript "Impact of color-coded and warning nutrition labelling schemes: a systematic review and network meta-analysis" (PMEDICINE-D-20-05962R3) for consideration at PLOS Medicine.

I have discussed the paper with editorial colleagues and it was also seen again by two reviewers. I am pleased to tell you that, provided the remaining editorial and production issues are dealt with, we expect to be able to accept the paper for publication in the journal.

[LINK]

Please let me know if you have any questions, and we look forward to receiving the revised manuscript.   

Sincerely,

Richard Turner, PhD

rturner@plos.org

Requests from Editors:

At line 9, please add a sentence, say, to state what the primary outcomes were.

Around line 11, we ask you to note the number or proportion of laboratory studies included (alternatively, this could be quoted as a limitation). 

At line 16, we think that "Nutri-score" (NS) needs to be spelt out at first use.

At line 24, we suggest "... content of purchases.".

At line 28, we suggest "... nudging consumers towards the purchase ..." or similar.

The sentence beginning at line 30 ("The difference could lie ...") is really a sentence of discussion, and we therefore ask you to remove it, and if you wish add a few words to this effect to the "Discussion" subsection of your abstract.

At line 34, we ask you to substitute "impact" in place of "effectiveness", bearing in mind that not all the evidence included is from randomized studies. 

At line 96, please make that "...had implemented".

At line 158, should that be "2021" as in the abstract?

At line 492, please make that "Despite the heterogeneity ...".

Throughout the text, please add a space immediate preceding reference call-outs (e.g., "... their products [7,8].".

Please remove the quotation marks from the title for reference 21.

Please spell out the "L" in the author of reference 45.

Comments from Reviewers:

Reviewer #1: 

[supportive report received]

Reviewer #4: 

The authors have addressed my second round of comments.

Michael Dewey

***

[LINK]

---

## [Editor Report · Decision Letter 4]

11 Aug 2021

Dear Dr He, 

On behalf of my colleagues and the Academic Editor, Dr Ares, I am pleased to inform you that we have agreed to publish your manuscript "Impact of color-coded and warning nutrition labelling schemes: a systematic review and network meta-analysis" (PMEDICINE-D-20-05962R4) in PLOS Medicine.

PRESS

Sincerely, 

Richard Turner, PhD 

rturner@plos.org